# Memorization Sinks: Isolating Memorization during LLM Training

**Gaurav R. Ghosal** [1]  **Pratyush Maini** [1,2]  **Aditi Raghunathan** [1]

## Abstract

Large language models are susceptible to memorizing repeated sequences, posing privacy and copyright concerns. A popular mitigation strategy is to remove memorized information from specific neurons post-hoc. However, such approaches have shown limited success so far. In a controlled setting, we show that the memorization of *natural* sequences (those that resemble linguistically plausible text) become *mechanistically entangled* with general language abilities, thereby becoming challenging to remove post-hoc. In this work, we put forward a new paradigm of `MemSinks` that promotes isolation of memorization by design. We leverage a sequence identifier to activate a unique set of memorization neurons for each sequence across repetitions. By analyzing the dynamics of learning and forgetting, we argue that `MemSinks` facilitates clean isolation of memorized content, making it easier to remove without compromising general language capabilities. We implement `MemSinks` at the billion-parameter and billion-token scale, and observe both effective isolation and strong generalization. To our knowledge, this is the first proof-of-concept on real data demonstrating that simultaneous generalization and isolation is achievable. We open-source our code at `http://github.com/grghosal/MemSinks`.

## 1. Introduction

Large language models often memorize sequences seen frequently throughout pretraining (Carlini et al., 2023; Nasr et al., 2023), posing concerns in privacy, copyright, and membership inference. There is significant research on the problem of "unlearning" or removing memorized infor-

mation from models post-hoc, including fine-tuning based approaches (Maini et al., 2022; Barbulescu & Triantafillou, 2024). However, presently all such methods present a substantial tradeoff between removing memorization and preserving general model capability.

We identify a core reason for the tradeoff: standard training induces mechanistic entanglement, where the same components support both generalization and memorization. This occurs when memorization relies on mechanisms also used for general language understanding. In a controlled setting, we show that natural-looking repeated sequences are memorized with strong entanglement, making post-hoc removal difficult without harming general performance. We further prove that gradient descent has an implicit bias toward such entangled solutions, suggesting that **mechanistic entanglement is inherent to current training methods.**

*Can new training approaches better disentangle memorization and general language capabilities?*

We begin with a natural disentanglement attempt (Section 4; see also (Cloud et al., 2024)): restricting gradient updates from repeated sequences to designated "memorized components", while the remaining "general components" learn only from non-repeated data. This approach has two major flaws. First, it weakens generalization by depriving general components of all training signal from potentially high-quality repeated sequences. Second, and more subtly, generalization further degrades when memorized components are removed at inference: the memorization components are active in the forward pass during training, causing the general components to implicitly rely on them. Their post-hoc removal breaks this dependence and harms performance.

Drawing lessons from the failures above, we introduce Memorization Sinks (`MemSinks`) — inspired by the previous study of localization in Maini et al. (2023).

> **Memorization Sinks** selectively activate sequence-specific memorization components alongside shared generalization components. Dropping memorization components erases the memorization of the corresponding sequence.

Note that this approach addresses the implicit dependence

---

[1]Department of Machine Learning, Carnegie Mellon University, Pittsburgh, Pennsylvania, USA [2]DataologyAI, Redwood City, CA, USA. Correspondence to: Gaurav Ghosal <gghosal@andrew.cmu.edu>.

*Proceedings of the $42^{nd}$ International Conference on Machine Learning*, Vancouver, Canada. PMLR 267, 2025. Copyright 2025 by the author(s).

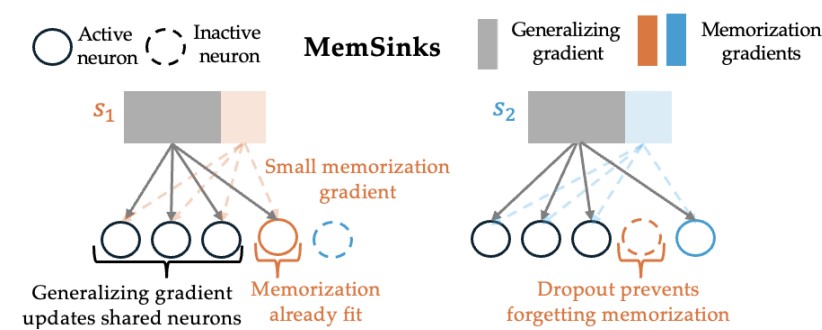

Figure 1. **Conceptual Intuition of `MemSinks`**. We conceptually partition the learning signal from each example to into "generalization" and "memorization" components. On the left, we show that standard training can store memorization signal in any neurons. In `MemSinks`, we provide a set of *memorization sink* neurons which are shielded from forgetting induced by other examples. As a result, (a) memorization accumulates in these neurons and (b) once the sink neurons fit memorization, it is no longer reinforced elsewhere in the model.

issue identified earlier, and also allows the shared components to learn from repeated sequences. But what prevents the shared components from containing memorized information? Does it suffer from the same mechanistic entanglement issue of standard training?

The magic lies in the difference in the training dynamics of generalizing and memorization signal, which has been studied in disparate lines of work. Generalizing signal is consistently amplified across training sequences (Chatterjee, 2020). However, memorization signals of different training sequences interfere with each other, resulting in a cyclical learning-forgetting dynamic (Toneva et al., 2018): learning when the repeated sequence is seen, but forgetting when training on a different sequence due to interference. In standard training, there is no separation of model components so this learning-forgetting dynamic occurs throughout the model leading to mechanistic entanglement. However, in `MemSinks`, this cycle is broken by allocating separate components per repeated sequence that are **protected from interference** with other sequences. As a result, memorized information is not reinforced in the shared components across repetitions of sequences.

How well does `MemSinks` disentangle memorization and generalization in practice? We train 360M and 1.7B SmolLM models (Allal et al., 2025) on the SlimPajama dataset (Shen et al., 2024), with repeated sequences drawn from TinyStories. Standard training leads to strong memorization where repeated sequences show much lower loss than held-out ones. With `MemSinks`, dropping the memorization components closes over 50% of this loss gap, mitigating memorization. Furthermore, `MemSinks` (without memorization components) matches the validation loss of standard training and significantly outperforms a deduplication baseline, preserving the benefits of repeated data for generalization. This provides a proof-of-concept

that `MemSinks` can disentangle memorization from generalization in realistic settings (Section 5.2).

We perform various ablations on the impact of design choices for implementing `MemSinks` on a small-scale TinyStories pre-training task in Sections 5.1 and 5.3. To assess the practicality of `MemSinks`, we investigate two key axes: (1) scalability with respect to model and data size, and (2) robustness to noise in sequence ID assignment. While our proof-of-concept assumes access to perfect sequence IDs, real-world deployments would require tolerance to imperfect or noisy estimates. We find that `MemSinks` is robust to small levels of noise in the sequence IDs (up to 10%) and works across a range of model sizes. Importantly, the benefits of `MemSinks` scale with increasing model size, suggesting promise in larger scales.

In summary, we argue that post-hoc unlearning on standard trained models is fundamentally limited due to inevitable mechanistic entanglement between memorization and generalization components. We propose a new training paradigm, `MemSinks`, and provide a conceptual explanation for how it enables clean disentanglement. Empirically, we show that `MemSinks` supports post-hoc removal of memorized content without compromising performance on real-world data. We also demonstrate its practicality: the approach scales well, with gains preserved, and even amplified, as we scale up models and datasets, and it remains robust under noisy sequence ID metadata. Ultimately, `MemSinks` offers a concrete path forward for the challenge of removing memorization without degrading model capabilities.

## 2. Related Works

**Forgetting Memorized Sequences.** With the discovery of memorization of sequences in large language models (Carlini et al., 2023; Nasr et al., 2023), there has been interest

in techniques to remove memorization in LLMs post-hoc. One class of such methods involves further training of *all* language model parameters in order to reduce the likelihood of a memorized sequence. For example Thudi et al. (2022) presents the simple technique of simply training to increase the loss of memorized examples. Liu et al. (2022) further regularizes this by concurrently minimizing the loss on a *retain set* of validation set examples. Other works have examined using preference-based training to incentivize the replacement of a memorized sequence with a safe alternative (Zhang et al., 2024; Maini et al., 2023). However, currently all such unlearning methods are prone to degrading general model capabilities, beyond the desired unlearning target (Maini et al., 2023). Moreover, they require the computational expense of retraining all weights and in many cases access to additional data in the form of a retain set.

**Localization of Memorization.** A promising class of methods for removing memorization seeks to *localize* specific model components, such as neurons, that are responsible for storing memorized sequences (Maini et al., 2023). Once such neurons are identified, they can simply be *dropped out* to ensure unlearning. These methods have the advantage of avoiding costly full model training and seek to avoid model degradation by minimizing the number of parameters changed. Localization techniques have achieved success in removing personally identifying information (PII) (Chen et al., 2024), as well as when there exist distinctive memorization triggers (Stoehr et al., 2024). However, all such methods present a tradeoff between removing memorization and preserving model performance (Chang et al., 2024b).

**Mixture-of-Expert Models.** Sparse mixture-of-experts models (S-MOE) have been studied in prior works as a means to expand model capacity without increasing compute costs (Shazeer et al., 2017). Like MemSinks, MOE models selectively activate fully-connected neurons depending on the input token. However, the selection of neurons in SMOE is typically learned subject to load-balancing constraints (as opposed to explicitly enforced as in MemSinks). Consequently, it is unclear what degree of localization is achieved by SMOE. Zoph et al. (2022) find that experts generally specialize based off of low-level syntactic structures rather than semantic features. Dai et al. (2024) find that there is often redundancy across experts in MOEs, challenging the idea that they reliably localize information within experts. Other works (Gururangan et al., 2021; Park et al., 2025) provide some evidence of localization in MOE models: however they measure localization relative to coarse attributes such as language and general topics. In this work we study the far more fine-grained problem of localizing memorization of specific documents or sequences.

## 3. The Pitfalls of Post-Hoc Localization

A substantial body of prior work seeks to remove memorized content from LLMs through post-hoc updates. These approaches attempt to precisely target the mechanisms responsible for memorization, while preserving general capabilities. Their success, however, hinges on the assumption that memorization and generalization are supported by distinct, non-overlapping mechanisms. In this section, we conduct controlled experiments to examine when such mechanistic separation holds. We compare two settings: (1) highly atypical canary sequences, commonly used in prior memorization studies, and (2) sequences resembling natural text from the pretraining corpus. We find that memorization of these more natural sequences is significantly harder to remove post-hoc, suggesting there may be **mechanistic entanglement** between memorization and general capabilities. In an analytical setting, we show that gradient descent actively prefers entangled solutions.

### 3.1. Experimental Setting

We train models on two controlled settings designed to induce different types of memorization: repeated natural text sequences and highly atypical canaries. We then test whether the memorized sequences can be removed from the model without inducing model degradation.

**Datasets.** We conduct our experiments in a controlled setting using a subset of the TinyStories dataset (Eldan & Li, 2023). In our first setting, we randomly sample 100 stories from the TinyStories training set and repeat them 128 times (TS-Repetition). Resultantly, memorized sequences are composed of natural text (coming from the same distribution as all other sequences). As a comparison, we study a second *canary-based* setting where memorized sequences are atypical (TS-Canary). We concatenate random sequences of tokens (Canaries) to 100 stories and repeat them 128 times in training. While TS-Canary more closely resembles memorization of label noise or atypical examples in supervised settings, it may be less representative of memorized documents in LLMs. In both cases, we additionally include 20,000 un-repeated TinyStories sequences (in total we train for ∼16M tokens).

**Evaluation Metrics.** We measure **sequence forgetting** as the difference in loss on repeated sequences before and after localization and dropout (higher is better). We measure the **model degradation** as the difference between the validation loss before and after removal (higher is better). This reflects that we hope to avoid increases in validation loss when removing memorization.

### 3.2. Empirical Observations

We show the results of our analysis in Figure 2. We observe that both post-hoc methods achieve limited success and struggle particularly to remove typical memorized sequences from `TS-Repetition`.

**Localization Methods Achieve Only Partial Success.** In Figure 2(a) we show the trade-off in sequence forgetting and model degradation of pruning. We observe in both settings that dropping out the identified neurons leads to an increase in the memorized sequence's loss, suggesting some success in localization. There are similar trends in Figure 2(b) for integrated gradients, although we observe it generally produces less model degradation than pruning. Additionally, we see that integrated gradients is less effective in removing memorization in `TS-Repetition`, while being highly effective in `TS-Canary`.

**Natural Sequence are Particularly Challenging to Remove.** Across both methods, we find that applying post hoc methods to `TS-Repetition` results in greater model degradation than `TS-Canary`. This difference is particularly pronounced for integrated gradients. Recall that the memorized sequences in `TS-Repetition` are natural – similar to the non-repeated training data and the validation set. Our results suggest the memorization of typical sequences may not be cleanly isolated from the model mechanisms responsible for general capabilities.

**No Clear Separation Between Natural Sequence Memorization and Generalization.** In Figure 2(c), we plot the validation and memorization of a model trained on `TS-Repetition`. We see that the loss on repeated sequences and the validation set descend simultaneously. The simultaneous learning of memorization and generalization seen here underscores the tight interplay between natural sequence memorization and general capabilities, contrasting form settings such as label noise where memorization occupies a distinct phase of training (Li et al., 2020).

Ultimately, our empirical results highlight that removing memorized natural text can be especially challenging for simple post-hoc methods. Next, we provide an explanation of this phenomena in a theoretical setup.

### 3.3. Theoretical Analysis of Mechanistic Entanglement

Our empirical results suggest that not all memorized sequences are equally easy to remove. While atypical canaries can be removed with minimal model degradation, removing natural memorized sequences tends to induce model degradation. Intuitively, this suggests that the memorization of natural text sequences is closely *entangled* with mechanisms responsible for general language modeling (thus being difficult to fully extricate post-hoc). In this section, we theoretically illustrate that such mechanistically entan-

gled solutions can be preferred by the training process.

We consider a setting where the model features are separated into a generalizing subspace $S_{\text{sem}}$ and a memorization subspace $S_{\text{mem}}$. We assume that memorization can be implemented either with an *entangled solution*, $\mathbf{W}_{\text{ent}}$ (reusing the features $S_{\text{mem}}$) or a disentangled solution $\mathbf{W}_{\text{dis}}$, using the orthogonal space $S_{\text{mem}}$. Our complete setting is described in Appendix D. Under some assumptions, we prove that gradient descent is biased towards reusing the features $S_{\text{sem}}$:

**Theorem 3.1** (*Informal*: Natural Sequence Memorization is Entangled)**.** *Consider training* $f(\mathbf{x}) = \mathbf{W}_{proj}\mathbf{W}_{fc}\mathbf{x}$ *on* $\mathcal{D}_{pre}$ *as in the setting of Appendix D. Then gradient flow does not converge to* $\mathbf{W}_{dis}$*.*

We defer the full proof to Appendix D, but discuss the intuition of our result here. Our result follows from past analyses of the minimum-norm bias of gradient flow. As a result of this bias, memorization perturbs the generalizing subspace and any post-hoc unlearning methods must also perturb $S_{\text{sem}}$. Had memorization been implemented with $S_{\text{noise}}$, unlearning could be accomplished through updates *orthogonal* to the general capabilities (reducing the risk of degrading general capabilities).

Ultimately, our findings in this section challenge the feasibility of relying on post-hoc approaches to remove memorized information. We show empirically and theoretically that memorization can be implemented closely entangled with general capabilities. In the remainder of the paper, we study ways to induce models to isolate the memorization of natural text sequences during training.

## 4. The Mirage of Forced Localization

In Section 3, we found that standard training techniques can result in memorization being difficult to decouple from the model's general capabilities. This suggests it may be necessary to specifically pre-train models for the ability to eliminate memorized information downstream.

Ideally, we would like to train models where sequence-specific memorization is handled by a distinct set of components (the *memorization component*), while general capabilities are encoded in a separate *generalization component*. This would enable removal of memorized information simply by modifying the memorization component, without risking any damage to the generalizing mechanisms. The most straightforward way to enforce this structure is by constraining the training gradients: updates from repeated, likely-to-be-memorized sequences can be restricted to the memorization component, while updates from all other sequences update the generalization component.

While the concept of forced localization is intuitively appealing, we empirically find two major drawbacks. First, it leads

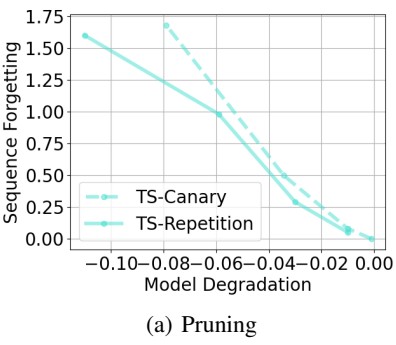

(a) Pruning

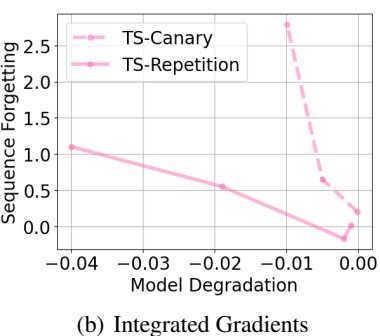

(b) Integrated Gradients

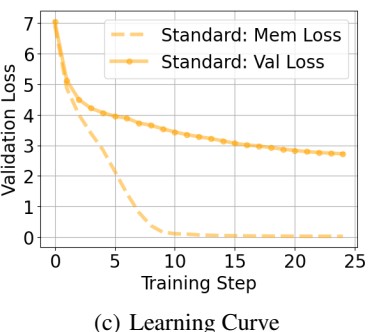

(c) Learning Curve

*Figure 2.* **Study of Localization** (a) We plot the unlearning-model degradation tradeoff of pruning by varying the number of dropped out neurons and demonstrate the method struggles to unlearn sequences of both kinds. (b) We plot the performance of integrated gradients and demonstrate that although it mitigates model degradation in both cases, it struggles with removing typical sequences. (c) Loss curve when training on `TS-Repetition`. We observe that memorization decreases alongside the validation loss, indicating that the model gains capability even as it memorizes sequences.

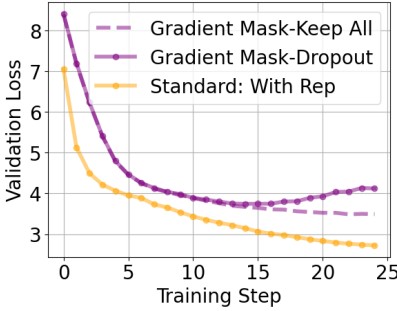

*Figure 3.* **Impact of Gradient-Masked Training on Validation Performance**. We compare the validation loss of gradient-masked training with (Gradient Mask-Dropout) and without (Gradient Mask-Keepall) memorization neurons removed to a standard training run (Standard: With Rep). We observe that (a) gradient-masked training learns slowly relative to standard training, achieving a significantly worse validation loss and (b) dropping out memorization neurons degrades validation performance as training progresses.

to significantly worse generalization compared to standard training. This suggests that shared components—those updated by gradients from all data points—are crucial for maximally learning general capabilities. Second, we find that removing memorization components still degrades model performance, indicating a failure to make memorization fully independent of the general capabilities.

## 4.1. Forcing Localization By Masking Gradients

We adopt a similar methodology to Cloud et al. (2024). In each layer we partition the intermediate neurons in the MLP into generalization and memorization neurons. During training, the gradients in the MLP layer from repeated sequences are masked to only modify weights corresponding to memorization neurons and conversely, non-repeated sequences

have their gradients routed to the generalization neurons. We provide complete experimental details in Appendix E.

## 4.2. Empirical Findings

**Forced Localization Impairs Generalization.** We observe that the performance of gradient masking is inferior to a standard model, even before memorization neurons are removed (Figure 3). This observation renders gradient masking impractical, as it significantly worsens the model's general capabilities. Our findings here highlight that some "shared" neurons – those that are updated by all sequences – are essential to maximally aggregate generalizing features. However, this suggests a potential tradeoff between generalization and disentangling memorization: shared neurons could entangle memorization with generalization.

**Localization Alone is Not Sufficient for Removal.** In Figure 3, we further compare whether the localization induced by Gradient Masking translates to distortion-free removal of memorization. Intuitively, if generalization and memorization are truly separated, it should be possible to remove memorization neurons without harming validation performance. In Figure 3 we examine the effect of removing memorization (seen in the gap between Gradient Mask-Keep All and Gradient Mask-Dropout). We observe that the validation loss becomes sensitive to the removal of memorization neurons as training progresses – dropping out the memorization component exhibits an increasing loss. This reveals, somewhat surprisingly, that localization alone does not guarantee degradation-free unlearning. In Section 4.3, we study this phenomena in an analytical setup.

## 4.3. Co-Adaptation of Model Components

Previously, we made an unexpected finding: segregating the memorization to specific neurons does not automatically

enable straightforward removal. In this section, we theoretically examine why this sensitivity arises by studying the training dynamics of a simplified gradient masking setup (described in Appendix F). We show a phenomena of *neuron co-adaptation* in which separately trained neurons can nevertheless become sensitive to each other's removal.

**Theorem 4.1** (*Informal*: Co-Adaptation between Model Components). *Consider training a two layer linear neural net $f$ with the forced-localization scheme in Appendix F. Let $\hat{f}$ denote a two layer linear neural net trained in a standard manner on $\mathcal{D}_{pre} \setminus (s^{mem}, y^{mem})$ and $f_d$ denote $f$ with memorization neuron dropped out, Then we have for a constant $c > 0$*

$$\left\| f(\mathbf{x})_d - \tilde{f}(\mathbf{x}) \right\|_2 \geq cN$$

Intuitively, the proof of Theorem 4.1 shows that although the memorized example is isolated to a memorization neuron, dropping out this neuron is insufficient to recover a model that was trained without seeing $(s^{mem}, y^{mem})$. Our proof relies on the fact that the forward pass activation of the memorization neuron affects the gradient update to the generalization neurons. Our analysis also predicts that this co-adaptation increases with additional training, mirroring the empirical results in Figure 3.

Taken together, our findings reveal that the intuitive idea of forcing localization of sequences falls short in crucial ways. Surprisingly, it is both too rigid—disrupting the model's ability to learn general features—and too permissive, allowing entanglement between memorization and general capabilities to persist. Can we overcome these opposing failure modes? In the next section, we show that a more subtle and targeted intervention in model training dynamics can unlock finer control over memorization and generalization.

## 5. Channeling Training Dynamics with `MemSinks`

In Section 4, we found that rigidly partitioning gradients to achieve disentanglement is insufficient. Our result suggest the need to enforce a more precise separation between memorization and general capability. However, it is challenging to exactly specify what consitutes memorization versus generalization during the training process.

To address this, we leverage the difference in training dynamics of memorization and generalization. During training, generalizing signals are reinforced across distinct sequences and are broadly amplified (Chatterjee, 2020). Sequence-specific memorization, on the other hand, experiences interference from other examples and is gradually forgotten. As a result, repeated sequences are cyclically learned and forgotten throughout the training process, thereby becoming defused throughout the model.

We propose to allow memorization to build up in pre-specified *memorization sink* neurons by shielding them from interfering updates. Intuitively, by setting aside a known and stable location for storing memorization, we reduce the need for it to be reinforced across all model parameters. After training, removal of memorization can be achieved by simply deleting the memorization sinks.

Our design requires two properties of the memorization sink neurons: (a) they must experience less interference from unrelated sequences and (b) they must avoid *co-adaptation* with the rest of the model (as discussed in Section 4). We propose that both objectives can be achieved through *sequence-dependent dropout*. Specifically, a different subset of sink neurons is activated for each input sequence. This selective activation reduces the frequency of interfering gradient updates to sink neurons, shielding them from forgetting. At the same time, their infrequent activation also regularizes the rest of the model against co-adaptation.

**Implementation.** We implement `MemSinks` in the hidden layer of transformer MLPs, as prior works have found MLPs play a crucial role in storing memorization (Nanda et al., 2023; Geva et al., 2021). We set a fraction $g$ of the hidden layer neurons in each layer as generalization neurons (which are activated across all sequences), while the remaining $1 - g$ fraction serve as memorization sink neurons. We assign each sequence in the pretraining dataset an ID and mask the memorization sink neurons deterministically as a function of the sequence ID. During all evaluations, we remove memorization sink neurons. Further implementation details are provided in Appendix G.

**Experimental Details.** We train a GPT Medium model (same as all previous experiments), where 70% of MLP neurons are shared and the remaining 30% are allocated to the pool of memorization neurons. We emphasize that there are *far less* memorization neurons than total sequences. Thus, we do not assume each sequence can be allocated its own memorization neurons. We set the memorization neuron dropout ratio $p$ to 0.3, but explore other choices in Section 5.3. We train on the `TS-Repetition` dataset from Section 3.

### 5.1. Validation of `MemSinks` in TinyStories

**`MemSinks` Enable Learning Across Sequences.** In Figure 4(a), we compare the validation loss of `MemSinks` with standard training with and without repeated documents. Firstly, note that standard training with repeated sequences outperforms filtering them out. This indicates that the model does learn general capabilities from observing documents repeated multiple times in our setting. Next, we compare the standard trained models with `MemSinks`. We observe that (evaluating without the memorization neurons), `MemSinks` achieves comparable validation loss to standard training.

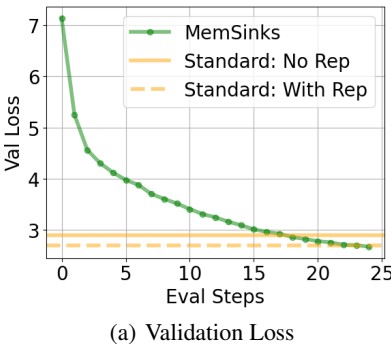
(a) Validation Loss

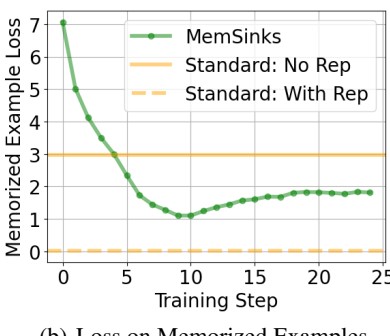
(b) Loss on Memorized Examples

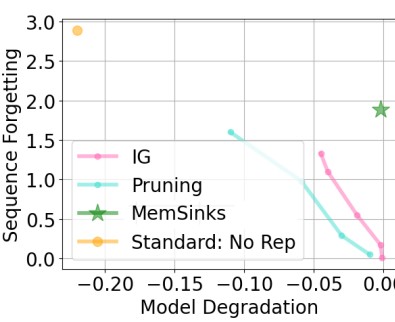
(c) Unlearning-Model Degradation Tradeoff

*Figure 4.* **Performance of `MemSinks`** (a) We find that `MemSinks` achieves a comparable validation loss to a normally trained model on `TS-Repetition`, outperforming a model trained without repeated sequences. (b) We show the loss of `MemSinks` on the repeated sequences, showing that it memorizes significantly less than a normally trained model. (c) We compare the sequence forgetting-model degradation tradeoff of `MemSinks`, relative to the post-hoc methods tested in Section 3, finding `MemSinks` outperforms both. We compute the model degradation for `MemSinks` and Standard: No Rep as the difference in the validation loss relative to a standard trained model on `TS-Repetition`.

**Dropping Out `MemSinks` Removes Memorization.** In Figure 4(b), we show the loss on the repeated TinyStories documents. A standard trained model memorizes these sequences during training, achieving close to 0 loss on them. We observe that dropping out the memorization neurons significantly increases the loss on these sequences, increasing the loss to roughly 66% of a standard trained model that does not memorize. Interestingly, we observe that in the latter part of training, the loss of sequence-tied dropout on memorized sequences begins to increase. This suggests that while shared neurons may initially implement some memorization, further training reverses this.

### 5.2. Larger Scale Experiments with `MemSinks`

In the previous section we studied `MemSinks` in a small scale setting. Now, we provide empirical validation of `MemSinks` on a larger scale, involving open pretraining datasets. We concentrate on settings where repeating data is beneficial for generalization (i.e. by upsampling a rare domain) but we also wish to avoid exactly memorizing the specific repeated documents.

**Setting.** We focus on pretraining SmolLM 360 and 1.7 billion models on 1 billion and 2 billion tokens, respectively. The majority of the pretraining dataset consists of a subset of the SlimPajama dataset, with a small amount being sampled from TinyStories. Our goal is to simultaneously improve performance on validation TinyStories data while also avoiding memorization of the repeated TinyStories training sequences. Full details are given in Appendix I.

**`MemSinks` Preserve Benefits of Repeated Data.** In the settings we study, validation performance on TinyStories benefits from repetition (by upweighting this relatively under-represented distribution). This can be seen in Figure

6(a) in the validation loss gap between the *deduplication* baseline (where the TinyStories are seen only once) and the full repetition baseline. We observe that the validation loss attained by `MemSinks` outperforms the deduplication baseline (achieving validation loss comparable to the repeated baseline). This provides evidence that `MemSinks` can leverage the generalization benefits of repeated data.

**`MemSinks` Mitigates Memorization.** Although repeating data has benefits for generalization, it also results in memorization – as evidenced by the gap between the validation loss and the loss on training examples. However, as shown in Figure 4(b) we find that training with `MemSinks` achieves a significantly higher loss on the training data, substantially closing the gap between validation and training loss (by at least 50%). In a setting with more extensive repetition (20x), we additionally find that the mitigation of memorization by `MemSinks` is more pronounced.

Collectively, our findings in this section provide promising proof-of-concept evidence that `MemSinks` can effectively separate memorization and generalization in the presence of heterogeneous and noisy data that constitute the real-world LLM pretraining corpora.

### 5.3. Practicality of `MemSinks`

In this section, we examine the robustness of `MemSinks` to various hyperparameters and variation in the training process. We focus on the loss on memorization examples as a measure of `MemSinks`'s success in isolating memorization. We perform the experiments here in the small-scale TinyStories setting from Section 5.1.

**Impact of `MemSinks` Activation Ratio.** A key hyperparameter for `MemSinks` is the fraction of memorization

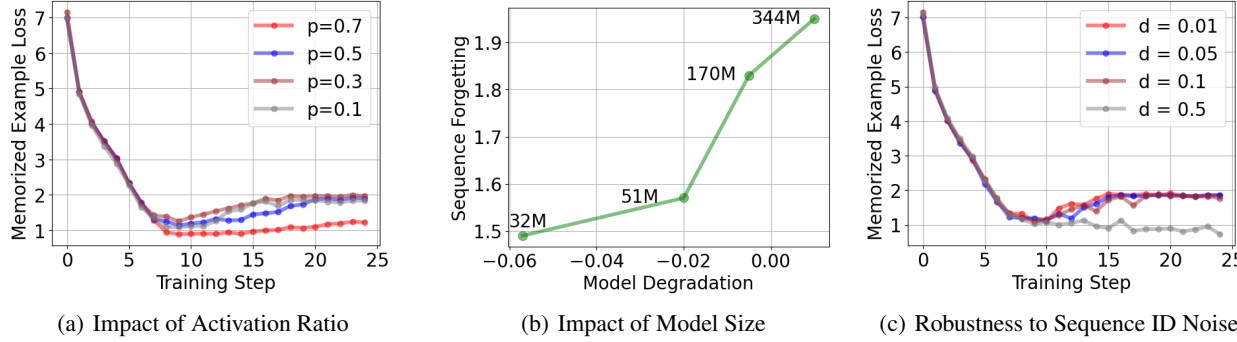

(a) Impact of Activation Ratio      (b) Impact of Model Size      (c) Robustness to Sequence ID Noise

*Figure 5.* **Practicality of `MemSinks`** (a) We study the impact of the fraction of memorization neurons activated ($p$) on any given sequence. We find that the model is generally robust to this choice, but activating too many can interfere in the isolation of memorization. (b) We plot the tradeoff between removing memorization and model degradation across model sizes, finding that larger models achieve a better tradeoff. (c) We study the impact sequence ID noise $d$, where a fraction of repeated documents have an inconsistent ID. We find that `MemSinks` withstands small amounts of noise (up to 10%).

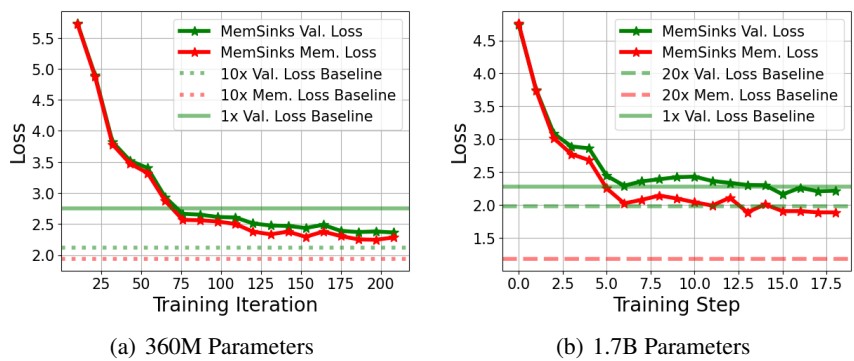

(a) 360M Parameters      (b) 1.7B Parameters

*Figure 6.* **Larger Scale Experiments on `MemSinks`** We plot the validation and memorization losses of (a) 360M and (b) 1.7B SmolLM-style models on a mixture of SlimPajama and TinyStories data. We compare with standard training with and without repeated Tiny Stories data. Our results demonstrate that across both settings, `MemSinks` is able to mitigate memorization (relative to standard training) without harming validation loss (outperforming or comparable to deduplicated training).

neurons activated on any given sequence ($p$). This controls across how many documents a particular memorization neuron is activated, as well as how much memorization capacity is allocated to a given sequence. We see that `MemSinks` is generally robust to this parameter, with a relatively small value $p = 0.3$ performing best. The performance of `MemSinks` does breaks down at higher levels (i.e. $p = 0.7$). We hypothesize this breakdown arises as a result of insufficient shielding of memorization sinks from forgetting and further investigate this in Section 5.4.

**Impact of Model Size.** Another concern is that `MemSinks` could necessitate using significantly larger models. In Figure 5(b), we test the performance of `MemSinks` on a range of model sizes and find that it is capable of isolating memorization across model scales—

as indicated by the comparably high losses on repeated sequences (relative to a normally trained model which attains nearly 0 loss). On the other hand, we find that model degradation (the increase in validation loss compared to a standard trained model of the same size) does grow as the model architecture becomes smaller. However, even on smaller models `MemSinks` outperforms post-hoc methods as shown in Figure 4(c). Thus, while model size plays a role in the success of `MemSinks`, the method has benefits in small models as well.

**Robustness to Masking Noise.** Implementing `MemSinks` requires that repetitions of the same sequence be presented with a consistent mask over the memorization sink neurons. Now, we investigate the robustness of `MemSinks` when this masking occurs inconsistently. Whenever a re-

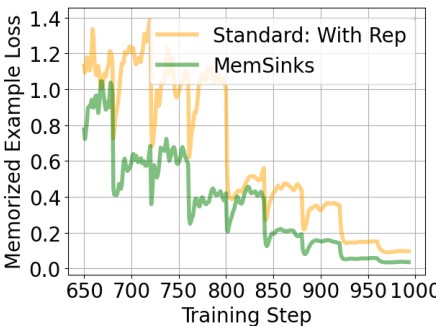

*Figure 7.* **Learning and Forgetting Dynamics of Standard Training and `MemSinks`.** We track the training loss on a single memorized example across the training trajectory for both standard training and `MemSinks`. We find that `MemSinks` experiences lower amplitude learning-forgetting cycles than standard training.

peated sequence is encountered, we randomly perturb its ID with probability $d$. Our results in Figure 5(c) show that `MemSinks` is robust to relatively small values of $d$, up to 10%. This suggests that having some amount of noise in the sequence IDs is permissible. On the other hand, when sequence IDs are highly inconsistent across repetitions (50% noise), `MemSinks` fails to isolate memorization as effectively. This verifies that consistency of sequence IDs across repetitions is an important factor behind its success.

### 5.4. Exploring the Mechanism of `MemSinks`

Recall that the motivation for `MemSinks`: to redirect sequence-specific memorization towards sink neurons. Earlier, we made an intuitive hypothesis that this could be accomplished by ensuring that these memorization sink neurons are activated less frequently. Here, we formalize this argument via empirical and theoretical investigations.

**Dynamics of Memorization Sink Neurons.** We first study the learning dynamics of the memorization sinks. We rerun our TinyStories pretraining setup with a single repeated sequence that is observed every 40 gradient steps. We track the training loss of this sequence over the course of training for both standard training and `MemSinks`. Recall that in `MemSinks`, the training loss on a sequence uses a forward pass with the shared neurons and the sequence's assigned memorization neurons activated. We track this training loss as it determines the size of the update experienced by the model from observing the sequence. We see that later in training, standard training continues to experience high-amplitude learning/forgetting cycles. `MemSinks`, on the other hand experiences less such fluctuations, maintaining a lower train loss on the repeated sequence. Our findings suggest that memorization sink neurons fit memorization and experience protection from forgetting.

**How Memorization Accumulates.** Previously, we saw

that sequence-specific memorization accumulates quickly in memorization sinks (relative to standard training). In Theorem H.2, we formally demonstrate that the amount of forgetting experienced on a given example depends on the number of steps taken on other examples, as well as the similarity of the other examples with the memorized example (measured in the cosine similarity). Based on this result, we show that the amount of memorization in the shared neurons can be upper bounded by a quantity dependent on the masking ratio, while it can be lower bounded in the memorization neurons. Our results theoretically mirror the trends seen in Figure 5(a).

## 6. Discussion

In this work, we have studied a crucial challenge in the responsible deployment of large-language models: removing memorized information without harming general model capabilities. Although extensive prior work has focused on *post-hoc* removal strategies, we highlight an important shortcoming of this approach: memorization can often leverage general capabilities – making degradation-free post-hoc removal impractical. Our findings argue for the need to more explicitly structure models to allow post-hoc unlearning to be performed reliably. Towards this paradigm, our work sheds light on important considerations for promoting this structure – highlighting why naive but intuitive approaches can be impractical. On the other hand, we uncover a framework for leveraging training dynamics to isolate memorization without tradeoffs in validation performance.

**Limitations.** As the focus of this paper has primarily been conceptual, much of the experiments are performed in small-scale and controlled settings. As such, further investigation is necessary to examine the impact of `MemSinks` on model capabilities that arise at larger scales. In addition, the current implementation of `MemSinks` crucially relies on meta-data which groups duplicated examples (so they can be given appropriate dropout masks). Although we provide evidence that `MemSinks` is robust to some level of inconsistency in these annotations, it is important to further investigate efficient techniques to generate this meta-data and how robust `MemSinks` remains in larger-scale settings. Finally, an important avenue for future work is verifying that `MemSinks` is robust to adversarial extraction techniques.

**Future Directions & Beyond Memorization.** Our work has primarily focused on verbatim memorization. However, changing the annotations used by `MemSinks` could allow for localization at various levels of granularity. For example, using domain or topic annotations could enable unlearning different data sources or topics. Future research can also examine the benefits of localization beyond unlearning. For example, better localization of facts could enable more reliable editing of knowledge.

## Impact Statement

This work aims to advance the field of Machine Learning by proposing a method to localize and manage memorized sequences in language models. Our approach, `MemSinks`, directly tackles a critical privacy concern: once a sequence is memorized by a model, unlearning it is difficult without extensive fine-tuning. By systematically isolating memorized information, our method enables safer, more targeted removal of private or sensitive text.

From an ethical perspective, the ability to better control and remove specific memorized sequences has potential societal benefits, particularly regarding user privacy and data protection. On the other hand, any mechanism that manipulates model internals could be misused for censorship or selective information removal if deployed irresponsibly. Mitigating such risks requires robust governance and transparent policies on what content may be unlearned.

Our proposal also contributes to the broader conversation about data governance in large-scale AI systems. As language models continue to grow in size and capability, balancing powerful generalization with privacy safeguards will become increasingly essential. We believe this work offers a practical step forward, but emphasize the need for interdisciplinary collaboration—spanning technical, ethical, and legal domains—to ensure that memory localization technologies support socially responsible AI systems.

## Acknowledgements

We gratefully acknowledge support from Apple, NSF and the AI2050 program at Schmidt Sciences. The authors would like to thank Jacob Springer, Chen Wu, Neil Kale, and Ziqian Zhong for their very helpful feedback and discussions during the onset of the project. Additionally, the authors are extremely grateful to Fahim Tajwar, Abitha Thankaraj, Tori Qiu, and Naveen Raman for their very helpful review and feedback on the finalized version of the paper. The authors are also very grateful to Sachin Goyal for his help and advice on using the LitGPT Library.

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

# A. Extended Related Works

**Memorization in Classification Settings.** A large body of work has focused on unlearning or forgetting memorized information from neural models, especially in the classification domain. This includes methods such as SISA (Bourtoule et al., 2021) that exactly unlearn information by maintaining multiple model copies, and a recent influx of approximate unlearning approaches (Triantafillou et al., 2023) that aim to perform post-hoc procedures on a model in order to remove information in question. The study of memorization in classification settings is connected to label noise and atypical examples (Harutyunyan et al., 2020; Feldman & Zhang, 2020). On the other hand, in this work, we concentrate on more natural or typical memorized points which often arise in the large language model setting.

# B. Implementation Details of TinyStories Training

**Implementation and Architecture.** We use the nanoGPT library to perform standard pretraining of the models. We train a GPT-2-Medium like architecture with embedding dimension 1024 and a 4 times expansion in the MLP layer. We used 24 layers, the resulting model had approximately 344 M parameters.

*Table 1.* Hyperparameter Tuning for Standard Training

| Parameter | Values |
|---|---|
| Max Learning Rate | {6e-5,6e-4,6e-3} |
| Weight Decay | {1e-5,1e-3,1e-1} |
| Min Learning Rate | $\frac{\text{Max Learning Rate}}{10}$ |
| LR Decay Steps | Total Training Steps |

**Hyperparameter Tuning.** We set the hyperparameters for our training as shown in Table 1. For parameters denoted in sets, we tuned over choices of these parameters relative to the validation loss. We also performed early stopping on the validation loss, but generally found that overfitting did not occur.

# C. Implementation Details of Post-hoc Localization Techniques

We generally follow the methodology used in (Chang et al., 2024b) and directly used their code as released online. We restrict our attention to their Hard-Concrete and Integrated Gradients methods presented in the papers.

**Hyperparameters: Hard Concrete.** We tuned $\lambda$, the $\ell_1$ loss coefficient used in training the pruning mask $M$ over the values $\{100, 500, 1000\}$ on a tuning set of 5 sequences. Additionally, we tuned the number of pruning iterations in the range $\{1000, 2000, 4000\}$. The remainder of hyperparameters were set to the optimal values reported by (Chang et al., 2024a). We tuned relative to the lowest validation loss achieved after dropping out the identified neurons.

**Hyperparameters: Integrated Gradients.** For Integrated Gradients, the only hyperparameter was the number of IG steps. As a result, we set this to the value reported in the paper, 16.

**Dropout Procedure.** Following the computation of mask scores by either Hard Concrete or attribution scores by Integrated Gradients, we sorted the neurons in each layer by these scores. Given a dropout parameter $r$, we dropped out an $r$ proportion of the neurons in each layer, as was performed in (Chang et al., 2024a).

# D. Disentanglement in Standard Training

**Setup.** We will consider a dataset with $n$ examples seen once $\mathcal{D}_{\text{once}} = \{(s_i, y_i)\}_{i=1}^{N}$ and a memorization sequence $(s^{\text{mem}}, y^{\text{mem}})$ that is repeated $k$ times. Thus, explicitly, our training dataset consists of $\mathcal{D}_{\text{pre}} = \mathcal{D}_{\text{once}} \cup (\cup_k (s^{\text{mem}}, y^{\text{mem}}))$. For the purposes of this theory, we will assume that the $\mathbb{R}^{d_{\text{emb}}}$ can be partitioned into a semantic subspace $S_{\text{semantic}}$ and a memorization subspace $S_{\text{noise}}$; explicitly we will write that $\phi(\mathbf{s^i}) = \begin{bmatrix} \phi(\mathbf{s})_{\text{mem}}^{(i)} & \phi(\mathbf{s})_{\text{mem}}^{(i)} \end{bmatrix}^{\top}$. We will make the assumption that $\forall i \in [1, N] \ (\phi(\mathbf{s})_{\text{mem}}^{(i)})^{\top} \phi(\mathbf{s})_{\text{mem}}^{(\text{mem})} = 0$. This can be interpreted as enabling us to uniquely identify a particular example in the noise subspace which would facilitate disentanglement of memorization. For convenience, we will consider the matrix implemented by $\mathbf{W} = \mathbf{W}_{\text{proj}} \mathbf{W}_{\text{fc}}$ and we will denote $\mathbf{W}^* = \text{argmin}_{L(\mathbf{W}, \mathcal{D}_{\text{once}})=0} \|W\|_F$ (i.e. the

minimum norm solution when $\mathbf{W}^*$ when the memorization example is not seen). We will assume that $\text{rank}(\mathbf{W}^*) > 2$. We will assume that we can achieve $0$ loss *using only the subspace $S_{\text{semantic}}$*. In particular, denoting the singular value decomposition of $\mathbf{W}^* = \mathbf{U}^* \Sigma (\mathbf{V}^*)^\top$, this translates to the condition that $\text{span}(\mathbf{V}^*) \subseteq S_{semantic}$. We will denote the term $\Delta_{\text{mem}} = y^{\text{mem}} - f(\phi(\mathbf{s}^{\text{mem}}))$ (the change in prediction necessary to memorize $s^{\text{mem}}$) and assume $||\Delta_{\text{mem}}|| > 1$. We will assume that all the singular values of $\sigma_i(\mathbf{W}^*) < \frac{1}{2k} \; \forall i \in [1, \text{rank}(\mathbf{W}^*)]$ (i.e. the generalizing solution has low norm) and that the memorization activations are unit norm $||(\phi(\mathbf{s})_{\text{mem}}^{\text{mem}}|| = 1)$. Finally, we will assume that the $\Delta_{\text{mem}} \perp \text{col}(\mathbf{W}^*)$.

**Desiderata of an Unlearnable Model** Intutively, the problem of unlearning is of efficiently finding a model which behaves as though it never saw the repeated example (i.e. recover a model such that $\mathbf{W} = \mathbf{W}^*$). However, we also wish to accomplish in ways that are efficient computationally and do not require significant access to $\mathcal{D}_{\text{pre}}$ (as the pretraining dataset can often be inaccessible in downstream updates). This precludes simply retraining the model on the set $\mathcal{D}_{\text{pre}} \setminus (s^{\text{mem}}, y^{\text{mem}})$. Intuitively, this is straightforward if the memorization of $(s^{\text{mem}}, y^{\text{mem}})$ makes use of the noise features in order to implement the memorization. Concretely, consider the solution $\mathbf{W} = \mathbf{W}^* + \Delta_{\text{mem}}(\phi(\mathbf{s})_{\text{mem}}^{\text{mem}})^\top$. This solution implements memorization in a *disentangled way* by isolating it in a direction (in input-space) that is completely orthogonal to the semantically meaningful, generalizing subspace. As a result, $(s^{\text{mem}}, y^{\text{mem}})$ can be unlearned using updates that have no effect on the predictions of any other sequence. This is becase we assume that $\forall i \in [1, N] \; (\phi(\mathbf{s})_{\text{mem}}^{(i)})^\top \phi(\mathbf{s})_{\text{mem}}^{(\text{mem})} = 0$ and thus for any unlearning update $\mathbf{u}\mathbf{v}^\top$ where $\mathbf{v} \in S_{\text{mem}}$, we have that $\forall i \; (\mathbf{W} + \mathbf{u}\mathbf{v}\top)\phi(\mathbf{s}^{(i)}) = \mathbf{W}\phi(\mathbf{s}^{(i)})$.

### D.1. Analysis of Natural Sequence Memorization

We will begin by describing the two possible solutions for memorizing

**Definition D.1** (Disentangled Memorizing Solution)**.** A learned parameter $\hat{\mathbf{W}}_{\text{dis}}$ implements the Disentangled Memorizing Solution if $\hat{\mathbf{W}}_{\text{dis}}$ can be written as $\mathbf{W}^* + c\mathbf{u}\mathbf{v}^\top$ where $\mathbf{v} \in S_{\text{sem}}$ and $\hat{\mathbf{W}}_{\text{dis}}$ achieves $0$ training loss on $\mathcal{D}_{\text{pre}}$

This definition follows along from our previous discussion of the desiderata of an unlearnable model. In particular, we constrain memorization to be stored using the orthogonal memorization component. This implies that the memorized sequence can be unlearned simply by taking gradient steps which do not impact the predictions on unrelated sequence. Hence, we are guaranteed not to distort the mechanisms responsible for the model's general capabilities.

Finally, we will discuss *entangled memorizing solutions*–those that require modifying the general capability subspace of the model and hence run the risk of incurring model degradation in subsequent unlearning.

**Definition D.2** (Entangled Memorizing Solution)**.** A learned parameter $\hat{\mathbf{W}}_{\text{ent}}$ implements the entangled memorizing solution if $\hat{\mathbf{W}}_{\text{ent}}$ achieves $0$ training loss on $\mathcal{D}_{\text{pre}}$ and $\hat{\mathbf{W}}_{\text{ent}}$ can be written as $\sum_{i=1}^{k} \sigma_i^{(\Delta)} \tilde{\mathbf{u}}_i \mathbf{v}_i^\top + \sigma_{k+1}^{\Delta} \tilde{\mathbf{u}}_{k+1} \mathbf{v}_{k+1}$ where the $\mathbf{v}_1, ... \mathbf{v}_i$ are the original right singular vectors of $\mathbf{W}^*$ and $\tilde{\mathbf{u}}_i$ are a potentially shifted set of singular vectors and $\tilde{\mathbf{u}}_{i+1}, \tilde{\mathbf{v}}_{i+1}$ are a new set of vectors and $\forall i \in [1, k]$ we have that $\sigma_i^{\Delta} \leq \sigma(\mathbf{W}^*)_i + \frac{||\Delta_{\text{mem}}||}{k+1}$ and $\sigma_{k+1}^{\Delta} \leq \frac{||\Delta_{\text{mem}}||}{k+1}$ where $k = \text{rank}(\mathbf{W}^*)$

Intuitively, entangled memorizing solution reuses and shifts vectors that are used by the generalizing solution. We impose the condition on the singular values to model that the implementation of memorization is "diffused" across multiple of the generalizing semantically meaningful features.

**Theorem D.3** (Memorization of Natural Sequences is Entangled)**.** . *Consider training a two layer linear network $f(x) = \mathbf{W}_{proj}\mathbf{W}_{fc}\mathbf{x}$ on $\mathcal{D}_{pre}$ with the squared loss. Suppose that $(s^{mem}, y^{mem})$ is natural. Let $\mathbf{W}^*_{mem}$ be the final function learned by gradient flow and $\mathbf{U}^*_{mem}\Sigma_{mem}(\mathbf{V}^*_{mem]})^\top$ be its singular value decomposition. Then, $\mathbf{W}^*_{mem} \neq \mathbf{W}_{dis}$.*

We will first provide some intuition for this statement. Intuitively, $f$ could either memorize $(s^{\text{mem}}, y^{\text{mem}})$ by (a) making use of the noise component of $\phi(\mathbf{s}^{\text{mem}})$ – thereby isolating the change in the model induced by memorization orthogonal to the generalizing features or (b) by implementing the memorization by shifting the generalizing features. We will consider the setting in which it is possible to implement memorization in both ways. However, unlearning memorization implemented with the generalizing features carries the risk of distorting general capabilities whereas memorization isolated to the orthogonal space can be easily and robustly removed.

As a result, the primary intuition behind Theorem D.3 is to demonstrate that the training dynamics will prefer more *harder to unlearn* solutions when memorizing natural sequences.

Before providing the proof of Theorem D.3, we will restate an important result on the implicit bias of gradient flow in two-layer neural networks from Varre et al. (2023). We will first specify a set initializations for which the result holds.

*Table 2.* Hyperparameter Tuning for Sequence-Tied Dropout

| Parameter | Values |
|-----------|--------|
| Max Learning Rate | $\{6e\text{-}5, 6e\text{-}4, 6e\text{-}3\}$ |
| Weight Decay | $\{1e\text{-}5, 1e\text{-}3, 1e\text{-}1\}$ |
| Min Learning Rate | $\frac{\text{Max Learning Rate}}{10}$ |
| LR Decay Steps | Total Training Steps |
| $g$ | $\{0.7, 0.9, 0.95\}$ |

**Definition D.4** (Orthogonal Feature Initialization). A two layer linear network $\mathbf{W}_{\text{proj}}\mathbf{W}_{\text{fc}}$ has orthogonal feature initialization if $\mathbf{W}_{\text{fc}} = \sqrt{2\gamma}\mathbf{P}$ where $\mathbf{P} \in \{\mathbb{R}^{d \times l} | \mathbf{P}\mathbf{P}^\top = \mathbf{I}\}$ for some $\gamma > 0$ and $\mathbf{W}_{\text{proj}} = 0$

**Theorem D.5.** *Consider training a two layer neural network parameterized by $(\mathbf{W}_{proj}, \mathbf{W}_{fc})$ with gradient flow and orthogonal feature initialization. Let $\beta = \mathbf{W}_{proj}\mathbf{W}_{fc}$. We will denote the set of optimal parameters for a given pretraining dataset $\mathcal{D}_{pre}$ as $I(\mathcal{D}_{pre})$*

1. *The parameters converge to a global optima (i.e. $\lim_{t \to \infty}(\mathbf{W}_{proj}(t), \mathbf{W}_{fc}(t)) \in I(\mathcal{D}_{pre})$)*

2. *The effective linear predictor converges to a min-norm solution, formally $\lim_{t \to \infty} \beta(t) = \beta^*$ where $\beta^* = argmin_{\beta \in I(D_{pre})} ||\beta||_F$*

Using the result from Theorem D.5, the proof of Theorem D.3 is straightforward and simply requires comparing the Frobenius norms of the disentangled and entangled memorizing solutions. In particular, because $||\phi(\mathbf{s}^{(i)\top})||_2 = 1$, we must have that $||\mathbf{W}_{\text{dis}}||_F^2 = ||\mathbf{W}^*||_F^2 + || ||\Delta_{\text{mem}}||_2^2$ (the remaining singular value comes from the orthogonal rank-1 term $\mathbf{u}\mathbf{v}^\top$). On the other hand, consider the entangled memorizing solution. We have that $||\mathbf{W}_{\text{ent}}||_F^2 \leq ||\mathbf{W}^*||_F^2 + 2\sum_{i=1}^{k} \sigma(\mathbf{W}^*)_i \frac{||\Delta_{\text{mem}}||}{k+1} + \frac{||\Delta_{\text{mem}}||^2}{k+1} \leq ||\mathbf{W}^*||_F^2 + 2\frac{||\Delta_{\text{mem}}||_2^2}{k} \leq ||\mathbf{W}^*||_F^2 + ||\Delta_{\text{mem}}||_2^2$

Hence $\mathbf{W}_{\text{dis}}$ cannot be the minimum norm solution and gradient flow will not converge to it.

## E. Implementation of Gradient Masking

We generally follow the implementation outlined in (Cloud et al., 2024). We partition each MLP layer into memorization and generalization neurons. We tune this delineation of memorization and generalization neurons by the proportion of generalization neurons $g$. We additionally partition our dataset into examples seen once and the repeated examples. During training, we mask the gradients in each MLP layer such that the gradients from the repeated examples update only a the memorization block, whereas gradients of all other examples are routed to the generalization block.

**Hyperparameter Tuning.** We show the hyperparameters tuned for this method in Table 2. Hyperparameter denoted in sets are tuned relative to the validation loss *before* dropping out memorization neurons.

## F. Analysis of Gradient Masking

**Data Distribution.** Like the previous analysis, we will consider a dataset of $N$ $(s_i, y_i)$ pairs which are only seen once, and a special $(s_{\text{mem}}, y_{\text{mem}})$ that is repeated in training and we wish to isolate. Thus, the total training set $D_{\text{pre}} = \{(s_i, y_i)\}_{i=1}^N \cup (s_{\text{mem}}, y_{\text{mem}})$.

**Model Structure.** We will use the same two layer neural network structure $f(x) = \mathbf{W}_{\text{proj}}\mathbf{W}_{\text{fc}}x$. However, for simplicity , we will consider only training the second layer (i.e. $\mathbf{W}_{\text{proj}}$. Further more, to implement gradient masking, we will partition the hidden space of the MLP into two components – a memorization component $\mathbf{W}_{\text{mem}}$ and a generalizing component $\mathbf{W}_{\text{gen}}$. In particular, we will assume the structure $\mathbf{W}_{\text{proj}} = \begin{bmatrix} \mathbf{W}_{\text{gen}} & \mathbf{W}_{\text{mem}} \end{bmatrix}$. Thus the prediction $f(x)$ could be written as $\mathbf{W}_{\text{gen}}(\mathbf{W}_{\text{fc}}x)_{\text{gen}} + \mathbf{W}_{\text{mem}}(\mathbf{W}_{\text{fc}}x)_{\text{mem}}$. Here, we use the shorthand $(\mathbf{W}_{\text{fc}}x)_{\text{gen}}$ to denote the entries of the hidden layer activations that are input to the generalization neurons, while $(\mathbf{W}_{\text{fc}}x)_{\text{mem}}$ denotes those that are input to the memorization neurons. We will denote the dropped out model $f_d$ as the model with the generalization neurons dropped (zero-ed) out. Concretely $f_d = \begin{bmatrix} \mathbf{W}_{\text{gen}} & \mathbf{0} \end{bmatrix} \mathbf{W}_{\text{fc}}\mathbf{x}$.

**Training Procedure.** We introduce the following simplified version of the gradient masking scheme we examine empirically. Concretely, we prescribe the following SGD-like update rules for $\mathbf{W}_{\text{gen}}$ and $\mathbf{W}_{\text{mem}}$, given a learning rate $\gamma$. To keep track of the parameter values at different time-steps, we will use a superscript.

$$\mathbf{W}_{\text{gen}}^{T+1} = \begin{cases} \mathbf{W}_{\text{gen}}^{T} + \gamma \nabla_{\mathbf{W}_{\text{gen}}^{T}} \mathcal{L}(f_\theta(x), y), & \text{if } x \neq s_{\text{mem}} \\ \mathbf{W}_{\text{gen}}^{T}, & \text{otherwise} \end{cases}$$

Similarly, we will consider the following for $\mathbf{W}_{\text{mem}}$:

$$\mathbf{W}_{\text{mem}}^{T+1} = \begin{cases} \mathbf{W}_{\text{mem}}^{T} + \gamma \nabla_{\mathbf{W}_{\text{mem}}^{T}} \mathcal{L}(f_\theta(x), y), & \text{if } x = s_{\text{mem}} \\ \mathbf{W}_{\text{mem}}^{T}, & \text{otherwise} \end{cases}$$

In both cases, consider $\mathcal{L}$ to be the squared loss function. Intuitively, $\mathbf{W}_{\text{mem}}$ receives gradient only from the $s_{\text{mem}}$ and *no other examples* whereas $\mathbf{W}_{\text{gen}}$ receives gradient from *all but* $s_{\text{mem}}$. Thus, by training construction, we could say that the mapping $(s_{\text{mem}}, y_{\text{mem}})$ is stored only in $\mathbf{W}_{\text{mem}}$ as its gradient cannot change any of the other parameters.

**Ground-Truth Unlearned Model.** When performing unlearning, we wish to obtain a model that behaves as if it had never observed $(s_{\text{mem}}, y_{\text{mem}})$. We denote such a model (trained under standard training with parameters $\gamma$ as $\hat{f}$).

**Initialization.** We assume that $\mathbf{W}_{\text{gen}}^{(0)} = 0$ and that $\mathbf{W}_{\text{mem}}^{(0)} = 0$. We consider $\mathbf{W}_{\text{fc}}$ is initialized at a non-zero value and does not change throughout training.

**Order of Observations.** We will assume that both $f$ and $\hat{f}$ observe the data points in the same order (i.e. $(s_1, y_1), ..., (s_{N_{\text{total}}}, y_{\text{total}})$. We will assume that $f$ observes $(s_{\text{mem}}, y_{\text{mem}})$ once at timestep $T$. With the setup in place, we will now introduce a lower bound in the difference between the predictions of a dropped out model $f_d$ and the *ground-truth* unlearned model $\hat{f}$. Concretely, we show:

**Theorem F.1** (Removing Memorization After Gradient Routing). *Suppose that we train $f$ on $\mathcal{D}_{LM} \cup (s_{mem}, y_{mem})$. Suppose that there have been $N$ gradient steps taken since the last time $\mathbf{W}_{mem}$ was updated. Additionally, assume that $\hat{f}$ is trained using the same hyperparameters but without observing $(\mathbf{s}^{mem}, y^{mem})$.*

$$\left\| f_d(\mathbf{x}) - \hat{f}(\mathbf{x}) \right\|_2 \geq (N)\gamma(1-\gamma)^N c^2 \|\mathbf{x}\|_2$$

*where $c = \min_{i,j} \mathbf{s}_i^\top \mathbf{s}_j > 0$.*

We now proceed with the proof:

**Proof** We will use the notation $\mathbf{W}_{\text{mem}}, \mathbf{W}_{\text{gen}}$ to denote the parameters of $f$ and $\tilde{\mathbf{W}}_{\text{mem}}, \tilde{\mathbf{W}}_{\text{gen}}$ to denote the parameters of $\tilde{f}$ (which does not observe the $(s_{\text{mem}}, y_{\text{mem}})$ but sees all other points in the same order).

Observe that, in the case of $f$, $\mathbf{W}_{\text{mem}}$ remains constant throughout the training process as does $\tilde{\mathbf{W}}_{\text{mem}} = 0$ by Equation F. Next, recall that by Equation F, we have that $\mathbf{W}_{\text{gen}}^T = \tilde{\mathbf{W}}_{\text{gen}}^{T-1}$. We then have the following.

$$\mathbf{W}_{\text{gen}}^{(T+N-1)} = \mathbf{W}_{\text{gen}}^{(T)} \prod_{j=1}^{N-1} (\mathbf{I} - \gamma \mathbf{s}_j \mathbf{s}_j^\top) + \gamma \sum_{k=1}^{N-1} \mathbf{y}_k \mathbf{s}_k^\top \prod_{j=k+1}^{N-1} (\mathbf{I} - \gamma \mathbf{s}_j \mathbf{s}_j^\top)$$

$$- \gamma \sum_{k=1}^{N-1} (\mathbf{W}_{\text{mem}} \mathbf{s}_k \mathbf{s}_k^\top) \prod_{j=k+1}^{N-1} (\mathbf{I} - \gamma \mathbf{y}_j \mathbf{y}_j^\top)$$

and

$$\tilde{\mathbf{W}}_{\text{gen}}^{(T+N-2)} = \tilde{\mathbf{W}}_{\text{gen}}^{(T-1)} \prod_{j=1}^{N-1} (\mathbf{I} - \gamma \mathbf{s}_j \mathbf{s}_j^\top) + \gamma \sum_{k=1}^{N-1} \mathbf{y}_k \mathbf{s}_k^\top \prod_{j=k+1}^{N-1} (\mathbf{I} - \gamma \mathbf{s}_j \mathbf{s}_j^\top)$$

Note that since $\mathbf{W}_{\text{all}}^T = \tilde{\mathbf{W}}_{\text{all}}^{T-1}$ and the order of observing data points is the same the parameter-space difference between $\mathbf{W}_{\text{all}}$ and $\tilde{\mathbf{W}}_{\text{all}}$ is calculated as

$$\mathbf{W}_{\text{gen}} - \tilde{\mathbf{W}}_{\text{gen}} = -\gamma \sum_{k=1}^{N-1} (\mathbf{W}_{\text{mem}} \mathbf{s}_k \mathbf{s}_k^\top) \prod_{j=k+1}^{N-1} (\mathbf{I} - \gamma \mathbf{s}_j \mathbf{s}_j^\top)$$

Next observe that

$$
\begin{aligned}
f_{\mathrm{d}}^{(T+n-1)}(\mathbf{x}) - \tilde{f}^{(T+n-2)}(\mathbf{x}) &= (\mathbf{W}_{\mathrm{gen}}^{(T+n-1)} + 0 * \mathbf{W}_{\mathrm{mem}}^{(T+n-1)})\mathbf{x} \\
&\quad - (\tilde{\mathbf{W}}_{\mathrm{gen}}^{(T+n-2)} + \tilde{\mathbf{W}}_{\mathrm{mem}}^{(T+n-2)})\mathbf{x} \\
&= (\mathbf{W}_{\mathrm{gen}}^{(T+n-1)} - \mathbf{W}_{\mathrm{gen}}^{(T+n-2)})\mathbf{x} \\
&= (-\gamma \sum_{k=1}^{N-1} (\mathbf{W}_{\mathrm{mem}}\mathbf{s}_k\mathbf{s}_k^\top) \prod_{j=k+1}^{N-1} (\mathbf{I} - \gamma\mathbf{s}_j\mathbf{s}_j^\top))\mathbf{x}
\end{aligned}
$$

Thus, we wish to lower bound

$$
\left\| (-\gamma \sum_{k=1}^{N-1} (\mathbf{W}_{\mathrm{mem}}\mathbf{s}_k\mathbf{s}_k^\top) \prod_{j=k+1}^{N-1} (\mathbf{I} - \gamma\mathbf{s}_j\mathbf{s}_j^\top))\mathbf{x} \right\|_2
$$

First we note that $\prod_{j=k+1}^{N-1}(\mathbf{I} - \gamma\mathbf{s}_j\mathbf{s}_j^\top))\mathbf{x} \in \mathrm{span}(\{\mathbf{s}_j\}_{j=1}^{N-1})$ and by the repeated application of variational characterization of eigenvalues $\left\|\prod_{j=k+1}^{N-1}(\mathbf{I} - \gamma\mathbf{s}_j\mathbf{s}_j^\top))\mathbf{x}\right\|_2 \geq (1-\gamma)^{N-1}\|\mathbf{x}\|_2$. Now, we will recall that $\mathbf{W}_{\mathrm{mem}} = \alpha\mathbf{y}_{mem}\mathbf{s}_{mem}^T$. We will use this to rewrite the above term as:

$$
|\gamma| \|\mathbf{s}_k\|_2 | \sum_{k=1}^{N-1} (\mathbf{s}_N^\top\mathbf{s}_k\mathbf{s}_k^\top \prod_{j=k+1}^{N-1} (\mathbf{I} - \gamma\mathbf{s}_j\mathbf{s}_j^\top)\mathbf{x})|
$$

Upon repeated application of the prior result and Lemma 2, we then have that

$$
\begin{aligned}
| \sum_{k=1}^{N-1} (\mathbf{s}_N^\top\mathbf{s}_k\mathbf{s}_k^\top \prod_{j=k+1}^{N-1} (\mathbf{I} - \gamma\mathbf{s}_j\mathbf{s}_j^\top)\mathbf{x})| &\geq | \sum_{k=1}^{N-1}(1-\gamma)^{N-1}\|\mathbf{x}\|_2 c^2| \\
&\geq \sum_{k=1}^{N-1}(1-\gamma)^{N-1}\|\mathbf{x}\|_2 c^2 \\
&\geq (N-1)(1-\gamma)^{N-1}\|\mathbf{x}\|_2 c^2
\end{aligned}
$$

by positivity of the summands we can eliminate the absolute value symbol in step 2 and we have used that for all $i \leq N-1$, $(1-\gamma)^i \geq (1-\gamma)^{N-1}$ which follows from $\gamma < 1$. This yields the desired lower bound.

**Lemma F.2** (Lower Bound of Outer Product in $\mathrm{span}(\{\mathbf{k}_1, ..., \mathbf{k}_n\})$). *Suppose* $\min_{i,j} \mathbf{k}_i^\top\mathbf{k}_j \geq c > 0$, $\mathbf{x} \in span(\{\mathbf{k}_1, ..., \mathbf{k_n}\})$, *and consider* $i, j \in [1, n]$. *Then*

$$
\left\|\mathbf{k}_i\mathbf{k}_j^\top\mathbf{x}\right\|_2 \geq c\|\mathbf{x}\|_2
$$

**Proof** $\mathbf{k}_j^\top\mathbf{x}$ is a scalar so we have that $\left\|\mathbf{k}_i\mathbf{k}_j^\top\mathbf{x}\right\|_2 = \|\mathbf{k}_i\| |\mathbf{k}_j^\top\mathbf{x}|$. Since $\mathbf{x}$ is in the span of $\{\mathbf{k}_i\}$, we have that $|\mathbf{k}_j^\top\mathbf{x}| \geq c\|\mathbf{x}\|_2$

## G. Implementation of `MemSinks`

**Model Architecture and Implementation.** We used the same model architecture as reported in Appendix B. We set the first $g$ fraction of neurons in each MLP as the "shared neurons" and left the remaining $1 - g$ fraction as the memorization neuron pool. We applied the dropout layer after the GeLU activation function, prior to the downprojection layer.

**Assignment of Sequence IDs.** We sequentially numbered the sequences in the TinyStories training set and use these indices as the sequence IDs.

**Hyperparameter Tuning.** In Table 3, we show the hyperparameter ranges tuned over for `MemSinks`. Hyperparameters denoted in sets were tuned over using the validation loss *when the memorization is dropped out*.

*Table 3.* Hyperparameter Tuning for Sequence-Tied Dropout

| Parameter | Values |
| --- | --- |
| Max Learning Rate | {6e-5,6e-4,6e-3} |
| Weight Decay | {1e-5,1e-3,1e-1} |
| Min Learning Rate | $\frac{\text{Max Learning Rate}}{10}$ |
| LR Decay Steps | Total Training Steps |
| $g$ | {0.1,0.3,0.5,0.7} |
| $p$ | {0.1,0.3,0.5,0.7} |

## H. Analysis of `MemSinks`

### H.1. Formalization of Training Process

**Architecture.** For simplicity, we study the training dynamics of an MLP layer $f(x) = \mathbf{W}_{\text{proj}}\mathbf{W}_{\text{fc}}\mathbf{x}$, where $\mathbf{W}_{\text{proj}} \in \mathbb{R}^{d_{\text{h}} \times d_{\text{emb}}}$, $\mathbf{W}_{\text{fc}} \in \mathbb{R}^{d_{\text{emb}} \times d_{\text{h}}}$. Here, $d_{\text{emb}}$ refers to the embedding size of the model and $d_{\text{h}}$ refers to the number of hidden neurons in the MLP. Given a sequence $\mathbf{s}$, we consider that $f$ takes in the final position embedding of $\mathbf{s}$, which we denote $\phi(\mathbf{s})$ and directly outputs the logits of the next token (i.e. $\text{softmax}(f(\phi(\mathbf{s})))$ is a probability distribution over the next token in sequence $\mathbf{s}$.

For convenience, we will denote the hidden activations of sequence $\mathbf{s}$ as $\mathbf{z}(\mathbf{s})$. In our analysis, we will assume that the activation space of $\mathbf{z}(\mathbf{s})$ can be split into two subspaces $\mathbf{z}(\mathbf{s}) = \begin{bmatrix} \mathbf{z}(\mathbf{s})_{\text{shared}} & \mathbf{z}(\mathbf{s})_{\text{mem}} \end{bmatrix}$. These components will correspond to our choice of shared and memorization neurons. We will additionally consider $\mathbf{W}_{\text{fc}}$ frozen throughout training and mainly study the training dynamics of $\mathbf{W}_{\text{proj}}$. Thus for convenience, we will also decompose $\mathbf{W}_{\text{fc}}$ into two column-blocks (corresponding to the shared and memorization neurons, respectively): $\mathbf{W}_{\text{proj}} = \begin{bmatrix} \mathbf{W}_{\text{proj}}^{\text{shared}} & \mathbf{W}_{\text{proj}}^{\text{mem}} \end{bmatrix}$

**Data Setup.** We will treat our data as (embedding, next token) pairs. We consider we have a repeated sequence $\mathbf{s}^{\text{mem}}$ with corresponding next token $\mathbf{e}^{\text{mem}}$. Next, we will assume we have a large dataset of sequences seen only once during training $\mathcal{D}_{\text{once}} = \{(\mathbf{s}^{(1)}, \mathbf{e}^{(1)}), ..., (\mathbf{s}^{(N)}, \mathbf{e}^{(N)})\}$. For simplicity, we will consider the case where $\forall i\ \mathbf{e}^{(i)} \neq \mathbf{e}^{\text{mem}}$. Since we treat $\mathbf{W}_{\text{proj}}$ as frozen, we will also define $\epsilon_{\text{shared}} = \min \mathbf{z}(\mathbf{s}^{\text{mem}})_{\text{shared}}^{\top}\mathbf{z}(\mathbf{s}^{(i)})_{\text{shared}}$ and likewise that $\epsilon_{\text{mem}} = \min_i \mathbf{z}(\mathbf{s}^{\text{mem}})_{\text{mem}}^{\top}\mathbf{z}(\mathbf{s}^{(i)})_{\text{mem}}$. Intuitively, these quantities lower bound how similar the activations in the shared and memorization neurons are between the repeated example and any other example. For simplicity we will assume that the $||\mathbf{z}^{(i)}||_2 = 1$ for all $\mathbf{z}^{(i)}$ and that the parameter $||\mathbf{W}_{\text{proj}}||_2 < \frac{C_{\text{proj}}}{2}$ remains bounded throughout training. Finally we assume that the ouput embeddings $e$ are mutually orthogonal.

**Training Process.** In standard training, we study the training trajectory (with learning rate $\gamma$) of minimizing the cross entropy loss with respect to the parameter $\mathbf{W}_{\text{proj}}$. We consider training with batch size 1.

### H.2. Forgetting Under Normal Training Dynamics

To begin, we introduce a result on the softmax with bounded inputs

**Theorem H.1** (Softmax on $\ell_\infty$ bounded vectors)**.** *Consider $x \in \mathbb{R}^d$ and suppose $\|x\|_\infty \leq C$. Then $\max_i(\sigma(x))_i \leq \frac{e^{2k}}{d-1}$ and $\min_i(\sigma(x))_i \geq \frac{e^{-2k}}{d}$*

*Proof.* $\sigma(x)_i = \frac{\exp(x_i)}{\sum_{j \in d}\exp(x_j)} \leq \frac{\exp(C)}{\exp(C)+(d-1)\exp(-C)} = \frac{\exp(2C)}{\exp(2C)+(d-1)} \leq \frac{\exp(2C)}{d-1}$. Likewise $\sigma(x)_i \geq \frac{\exp(-C)}{\exp(-C)+(d-1)\exp(C)} = \frac{\exp(-2C)}{\exp(-2C)+(d-1)} \geq \frac{\exp(-2C)}{d}$. $\square$

Given our assumption that $||\mathbf{z}^{(i)}||_2 = 1$ and the bounded parameter norm assumption $||\mathbf{W}_{\text{proj}}||_2 < \frac{C_{\text{proj}}}{2}$, it follows that $||\mathbf{W}_{\text{proj}}\mathbf{z}^{(i)}||_\infty \leq \frac{C_V}{2}$. By Theorem H.1, we have that the entries of $\frac{\exp(-C_{\text{proj}})}{d_{\text{emb}}} \leq \sigma(f(\mathbf{z}^{(i)}) \leq \frac{\exp(C_{\text{proj}})}{d_{\text{emb}}-1}$, element wise. In the remainder of the theory, we denote $c_{min} = \frac{\exp(-C_{\text{proj}})}{d_{\text{emb}}}$ and $c_{max} = \frac{\exp(C_{\text{proj}})}{d_{\text{emb}}-1}$.

We will first show that the memorization of the repeated sequence $\mathbf{s}_{\text{mem}}$ is forgotten when we take intervening steps on non-repeated sequences $\mathbf{x}\mathbf{s}^{(i)}, ..., \mathbf{s}^{(i+n)}$. Formally, we have the following proposition. Formally, suppose that at after step $i$,

we have just seen $\mathbf{s}^{(mem)}$. Then we will show that the logit $\mathbf{e}^{(mem)}$ decreases during subsequent training steps $i$ through $i + n$. For this analysis, we will focus on the dynamics the shared neurons.

**Theorem H.2** (Forgetting in Standard Training). *Suppose we take a gradient step on* $\mathbf{s}^{(mem)}$ *at gradient step* $i$ *and subsequently make gradient updates on non-repeated sequences* $\mathbf{s}^{(i)}, ..., \mathbf{s}^{(i+m)}$. *After the* $m$ *gradient steps, we have that* $(\mathbf{e}^{mem})^\top f^{(i+m)}(\mathbf{z}^{mem}) \leq (\mathbf{e}^{mem})^\top f^{(i)}(\mathbf{z}^{mem}) - \gamma m \epsilon c_{min}$.

*Proof.* Only the parameter $\mathbf{W}_{\text{proj}}$ changes throughout training, so we can restrict our attention to its dynamics. We have that the gradient of $\mathbf{W}_{\text{proj}}$ on the sequence-next token pair $(\mathbf{z}, \mathbf{e})$

$$\frac{\partial L}{\partial \mathbf{W}_{\text{proj}}} = (\mathbf{e} - \sigma(f(\mathbf{z}))\mathbf{z}^\top$$

Now let $\mathbf{W}_{\text{proj}}^{(i)}$ denote the parameter value of $\mathbf{W}_{\text{proj}}$ after the $i$-th observation. We have that

$$\mathbf{W}_{\text{proj}}^{(i+m)} = \mathbf{W}_{\text{proj}}^{(i)} + \gamma \sum_{j=1}^{m} (\mathbf{e}^{(j)} - \sigma(f^{(j+i)}(\mathbf{z}^{(\mathbf{i})})\mathbf{z}^{(\mathbf{i})}{}^\top \tag{1}$$

where we will denote $f^{(j+i)}$ as the model with parameter $\mathbf{W}_{\text{proj}}$. Then, we have that the logit on the correct next token for memorized example $\mathbf{z}^{mem}$ is

$$(\mathbf{e}^{mem})^\top f^{(i+m)}(\mathbf{z}^{mem}) = (\mathbf{e}^{mem})^\top f^{(i)}(\mathbf{z}^{mem}) + (\mathbf{e}^{mem})^\top \gamma \sum_{j=1}^{m} (\mathbf{e}^{(j)} - (\mathbf{z}^{mem})\sigma(f^{(j+i)}(\mathbf{z}^{(\mathbf{i})})\mathbf{z}^{(\mathbf{i})}{}^\top (\mathbf{z}^{mem})$$

Now, since we have that the token embeddings are orthogonal, we can rewrite this as

$$(\mathbf{e}^{mem})^\top f^{(i+m)}(\mathbf{z}^{mem}) = (\mathbf{e}^{mem})^\top f^{(i)}(\mathbf{z}^{mem}) - (\mathbf{e}^{mem})^\top \gamma \sum_{j=1}^{m} \sigma(f^{(j+i)}(\mathbf{z}^{(\mathbf{i})})\mathbf{z}^{(\mathbf{i})}{}^\top (\mathbf{z}^{mem})$$

Note that by the assumption of bounded norm for $\mathbf{W}_{\text{proj}}$. we have that $(\mathbf{e}^{mem})^\top \sigma(f^{(j+i)}(\mathbf{z}^{(\mathbf{i})})) \geq c_{min}$ (defined earlier). Note also the assumption that $\mathbf{z}^{(\mathbf{i})}{}^\top (\mathbf{z}^{mem}) \geq \epsilon \ \forall i$. This implies that

$$(\mathbf{e}^{mem})^\top f^{(i+m)}(\mathbf{z}^{mem}) \leq (\mathbf{e}^{mem})^\top f^{(i)}(\mathbf{z}^{mem}) - \gamma \sum_{j=1}^{m} \epsilon c_{min} \tag{2}$$

This immediately yields our desired claim. $\square$

Next, we will show that the seqTD accumulates memorization in the memorization neurons, as formalized in the following theorem. This theorem also crystalizes some key quantities relating to gradient interference. First of all, we see that the forgetting depends on the number of *further gradient steps* taken after seeing $\mathbf{s}^{mem}$. Secondly, we observe that the impact of forgetting dynamics is influnced by how *similar* the activation of neurons are amongst different examples: controlled by $\epsilon$. The first observation immediately suggests that if some neurons were activated less often, then those neurons would be effectively "store" more memorization.

### H.3. Analysis of `MemSinks`

**Theorem H.3** (`MemSinks` Accumulates Memorization in Memorization Neurons). *Consider training* `MemSinks`, *where the memorization neurons are activated on a* $p$ *fraction of non-repeated examples. We will assume that the model is trained from 0 initialization. Denote the MLP* $f_{mem\text{-}dropped}$ *as the model with memorization neurons dropped out and* $f_{gen\text{-}dropped}$ *as the model with the generalization neurons dropped out. Suppose that the model is trained for* $N$ *total steps and the repeated sequence* $\mathbf{s}^{mem}$ *is observed* $k$ *times. Then we have at the end of training*

1. $(\mathbf{e}^{mem})^\top f_{gen\text{-}only}^{(n)}(\phi(\mathbf{s}^{mem})) \leq \gamma k(1 - c_{min}) - \gamma(N - k)\epsilon_{shared}c_{min}$

2. $(\mathbf{e}^{mem})^\top f_{mem\text{-}only}^{(n)}(\phi(\mathbf{s}^{mem})) \geq \gamma k(1 - c_{max}) - \gamma(N - k)\rho\epsilon_{mem}c_{max}$

*where $c_{min}$ and $c_{max}$ are constants depending on an upper bound of the parameter norm of $\mathbf{W}_{proj}$.*

*Proof.* Our argument resembles the proof of Theorem H.2, and we will rely on the intuition therein. For reference, we will write the gradients for the components of $\mathbf{W}_{\text{proj}}$ below.

$$\frac{\partial L}{\partial \mathbf{W}_{\text{proj}}^{\text{shared}}} = (\mathbf{e} - \sigma(f(\mathbf{z}))\mathbf{z}_{\text{shared}}^\top$$

and likewise

$$\frac{\partial L}{\partial \mathbf{W}_{\text{proj}}^{\text{mem}}} = (\mathbf{e} - \sigma(f(\mathbf{z}))\mathbf{z}_{\text{mem}}^\top$$

We will first examine $(\mathbf{e}^{\text{mem}})^\top f_{gen-only}^{(n)}(\mathbf{z}^{\text{mem}})$. At any point in training, recall that we can upper and lower bound the value $c_{min} \leq (\mathbf{e}^{(i)})^\top \sigma(f(\mathbf{z}^{\text{mem}})) \leq c_{max}$. As such, observe that $(\mathbf{e}^{(i)})^\top \sigma(f(\mathbf{z}^{\text{mem}}))$ received $k$ updates upper bounded by $\gamma(1 - c_{min})$ (from the $k$ obervations of $\mathbf{z}^{\text{mem}}$ and $(N - k)$ updates upper bounded by $\gamma\epsilon_{shared}c_{min}$ (from the remaining $(N - k)$ observations of the $\mathbf{z}^{(i)}$. This yields the desired claim for (1).

Now, for claim (2) observe that the component $(\mathbf{e}^{\text{mem}})^\top f_{mem-only}^{(n)}(\mathbf{z}^{\text{mem}})$ receives $k$ updates lower bounded by $(1 - c_{max})$ (again, from the $k$ observations of $\mathbf{z}^{\text{mem}}$, but only $p(N - k)$ updates from other observations, which can likewise be lower bounded by $\gamma\epsilon_{mem}c_{max}$ This immediately implies the desired claim in (2) $\qquad\square$

This theorem formalizes the notion that memorization "accumulates" in the memorization neurons when they are shielded from the interference of other sequences sufficiently. In our theory, the extent to which this occurs is dependent on two quantities (1) the fraction of *non-repeated* sequences for which the memorization neurons are active and (2) the similarity of activations of the repeated example and non-repeated example in the memorization neurons. Relative to algorithm design, however, we will generally only have control over $\rho$ and so we will consider $\epsilon_{\text{shared}} = \epsilon_{\text{mem}}$ out of convenience. Our analysis demonstrates that when $\rho$ is set appropriately low. Some calculation demonstrates that when $\rho < \frac{c_{min}}{c_{max}} - \frac{k}{(N-k)}(c_{max} - c_{min})$, then we will have a seperation in the logits of $\mathbf{s}^{\text{mem}}$ where the memorization neurons primarily contain the memorized example.

## I. Large Scale Implementation Details

Here, we report all experimental and implementation details for the large-scale experiments conducted in this paper

**Pretraining Data Mixture.** We consider pretraining datasets of 1 and 2 billion parameters (respectively for the 360M model and 1.7B model). The majority of pretraining tokens are sourced from a subset of the SlimPajama dataset (Shen et al., 2024). However, we consider a setting in which we have an *under-sampled* domain of 5000 TinyStories examples. We ensure that these 5000 TinyStories (Eldan & Li, 2023) documents are included in the pretraining data before filling the remaining tokens from SlimPajamas.

*Table 4.* Hyperparameters for Standard Training (360M)

| Parameter | Values |
|---|---|
| Weight Decay | 0.1 |
| Max Learning Rate | 5e-4 |
| Min Learning Rate | $\frac{\text{Max Learning Rate}}{10}$ |
| LR Decay Steps | Total Training Steps |
| Batch Size | 1024 |

*Table 5.* Hyperparameters for Standard Training (1.7B)

| Parameter | Values |
|---|---|
| Weight Decay | 0.1 |
| Max Learning Rate | 5e-5 |
| Min Learning Rate | $\frac{\text{Max Learning Rate}}{10}$ |
| LR Decay Steps | Total Training Steps |
| Batch Size | 1024 |

**Model Architectures.** We pretrain using SmolLM family models of size 360M and 1.7B. For standard training runs, we do not modify the architecture in any way. We describe the modifications necessary for `MemSinks` below.

**Training Hyperparameters.** We report the hyperparameters used for training the models (at size 360M and 1.7B, respectively) in Tables 4 and 5. For fairness, we use the same hyperparameters to train `MemSinks`.

### I.1. Implementation of `MemSinks`

We will release a full implementation of `MemSinks` at http://github.com/grghosal/MemSinks. Our implementation builds off of the LitGPT library (AI, 2023).

**Assignment of Sequence IDs.** We assign each document (example) in the pretraining corpus with a sequence ID that is a *hash* of the tokens within that document. We implemented a tensorized hash function in pytorch which will be released with our code.

**Tokenization and Packing.** When implementing `MemSinks`, it is necessary to input the sequence identification meta-data into the forward pass of the transformer model. In order to accomplish this while still leveraging the optimized streaming data loaders typically used in pretraining, we *interleaved* the sequence IDs into the stream of tokens saved during the tokenization step. That is, the pretraining data stream contained tokens at every even position and the previous tokens ID code at every odd position. This enables seamless loading of the sequence ID data alongside the tokens during training. During training, the sequence ID data and tokens are separated using a simple indexing operation. As opposed to the smaller-scale TinyStories setting, we pass a sequence ID with *each token* to enable training across document boundaries and prevent the necessity of padding tokens.

**Online Computation of `MemSinks` Masks.** Another crucial challenge faced when scaling `MemSinks` is the need to deterministically generate a neuron mask for each token in the sequence. While we used the standard Pytorch seeded random number generator during our smaller-scale experiments, this becomes impractical when different tokens in a batch can have separate masks. As Pytorch does not support batched-seeding in its random number generator, we implemented a tensorized linear congruential random number generator to generate neuron masks online during training. This enabled us to avoid needing to precompute masks across a 1 billion token training set.

**Placement of `MemSinks`.** We implemented `MemSinks` at every hidden MLP layer by partitioning the neurons into generalizing and memorization sink groups. We then apply the mask over the memorization neurons. For efficiency, we used the same masking across all MLP layers. As the SmolLM (Allal et al., 2025) family models uses a GatedMLP implementation, we apply `MemSinks` in the output-space after the gating.

**`MemSinks` Hyperparameters.** For `MemSinks`, for both model sizes, we tune across hyperparameter choices given in

*Table 6.* Hyperparameters for `MemSinks` (both sizes)

| Parameter | Values |
|---|---|
| $1 - g$ | $\{0.05, 0.1, 0.3, 0.5\}$ |
| $p$ | $\{0.1, 0.3, 0.5\}$ |

Table 6.

