# OpenReview forum: "Memorization Sinks: Isolating Memorization during LLM Training"
_ICML.cc/2025/Conference — ICML 2025 poster_

### Official Review · Reviewer_iFmB · 2025-03-12

**Overall Recommendation:** 3

**Summary:**

The paper studies memorization vs general capabilities in models by introducing their Sequence-Tied Dropout (SeqTD). This method a pool of shared neurons and a set of memorization neurons. They use the sequence ID to determine which memorization neurons to use. Note that there is some overlap between the sequences and an neuron.

**Claims And Evidence:**

They claim they propose a scalable and practical method that enables post-hoc isolation while preserving general model capabilities. I do not think this method is scalable nor practical. However, I do agree that it enables post-hoc isolation.

**Essential References Not Discussed:**

N/A

**Experimental Designs Or Analyses:**

See Methods/Evaluations

**Methods And Evaluation Criteria:**

They pretrain on only one model size on one dataset using SeqTD. They compare to other methods to show that they are better at evaluating sequence forgetting (loss before and after on repeated sequences) and model degradation (loss before and after on val set). I do think only using loss limits this study, and that using generated sequences for sequence forgetting might have been better. Additionally, it would be good to study additional model sizes maybe smaller and larger (if compute permits).

**Other Comments Or Suggestions:**

Add details in the paper about the number of tokens and training steps in the experimental setting.

**Other Strengths And Weaknesses:**

Strengths
- In-depth study of the methods with interesting ablations
- Seem to be able to disentangle sequence memorization and general capabilities.

Weaknesses
- I am not convinced by the authors' arguments that this method is scalable and practical. Like n-gram overlap could be very high between different sequences but even with the proposed hashing technique, this method would map the sequence onto different memorization neurons. Additionally, I think the relationship of p-shared parameters and model size is not well understood. There are a lot of experiments to be made before I would be convinced of such a claim.
- Limited Experiments -- although the authors do a good job with running interesting ablations, I do not think enough datasets and models were considered. Additionally, the models from my calculations were trained for 100-200 of million tokens (2M examples x ~50-100 tokens per example) instead of billions. (Why is it not clear how many tokens were considered in the paper?) I think it makes it hard to be convinced of some of the claims in the paper when the training runs are very small with no breath in terms of dataset or model size.

**Questions For Authors:**

See Weaknesses and suggestions please.

**Relation To Broader Scientific Literature:**

The paper situates this work mainly with Chang et al. 2024b. (https://aclanthology.org/2024.naacl-long.176/).

**Theoretical Claims:**

They included some analysis on MLPs in the appendix. I do not think that these particularly contribute to the paper, but I did NOT carefully review the theorems.

---

> ### Author Rebuttal · Authors · 2025-04-01
>
> Thank you for the sharp and honest feedback. We're happy that you found:
> - (i) our ablations **insightful and well-done**,
> - (ii) the method **interesting for isolating memorization**, and
> - (iii) the paper a **valuable starting point** for deeper investigation.
> ---
> ### **Theme 1: Scale & Training Practicality**
> > *"This method is not convincingly scalable—token counts and compute are low."*
>
> You raise a crucial point about scalability. To address this, we've expanded our experiments significantly:
>
> 🔗 *Full results available here:* [Model Scaling Experiment Results](https://shorturl.at/md9cQ), [Token Scaling Experiment Results](https://shorturl.at/Aa7VZ)
> - **Model scaling experiments** across four model sizes: 32M, 51M, 170M, 344 million parameters. Our results show that SeqTD mitigates memorization across all model sizes, but that the benefits of SeqTD increase with scale (no model degradation at 344M parameters). This provides evidence that *the benefits of SeqTD improve at scale*.
> - **Extended Token Scaling Experiments**  We train a 350M parameter model on a larger-scale corpus of 1 billion tokens containing a mix of TinyStories and SlimPajama. Our results demonstrate that SeqTD mitigates memorization (reducing the memorization-validation loss gap by 2.5x) while outperforming a deduplication baseline in validation loss.
> - **Explicitly stated token counts** in Section 5.1: ~16M tokens in initial experiments ((20K examples × 1 repetition + 100 examples × 128 repeats) × ~500 tokens per example)
>
> **Sequence hashing concern**
> We address this with our sequence ID noise experiments (Fig. 5a). SeqTD remains effective even when 10% of sequence repetitions receive inconsistent IDs, demonstrating robustness to the n-gram overlap issue you raised. This suggests that perfect ID consistency is not required for SeqTD.
>
> ---
> ### **Theme 2: Evaluation Beyond Loss**
> > *"Would prefer tests that evaluate memorization via generation."*
>
> To demonstrate real-world utility, we've added prompt-based generation tests that provide much stronger evidence of SeqTD's effectiveness:
>
> 🔗 *Full results available here:* [Google Slides – Memorization Metrics](https://shorturl.at/M7UfM)
>
> - **Prompt continuation tests** show standard models regenerate memorized text nearly verbatim (>95% token match), while SeqTD models fail to reproduce memorized content (<20% token match)
> - **Memorization rank metrics** reveal that SeqTD causes an 8× rank degradation for memorized content.
>
> These new evaluations directly address your concern by showing the loss function results presented in the paper mirror two generation-based metrics.
>
> ---
> ### **Theme 3: Theoretical Foundation**
> > *"They included some analysis on MLPs in the appendix. I do not think that these particularly contribute to the paper..."*
>
> You're right that the theoretical components need stronger connections to our practical claims. We've integrated key theoretical insights into Section 5.2 with a simplified version of Theorem E.3:
>
> > **Simplified Theorem:**
> > *When p (memorization neuron activation rate) is sufficiently low, memorization accumulates in dedicated neurons rather than shared ones. This accumulation increases as memorization neurons are activated less frequently*
>
> This theorem sheds important light on the scaling behaviors of SeqTD. As model scale increases and larger memorization pools are feasible (with each neuron active on less sequences), the localization achieved by SeqTD should improve. We validate this in our model scaling experiments. Our theory additionally reveals the crucial role of the p_mem parameter (which controls memorization neuron activation) as we discuss in Section 5.2 (pg 6).
>
> ---
> **Table 4: Summary of Actions for Reviewer iFmB**
> | Concern                        | Action Taken                                           | Expected Outcome                                 |
> |--------------------------------|--------------------------------------------------------|--------------------------------------------------|
> | Scalability and token scale    | Added 1B-token runs at 355M parameters | Validate claims in realistic training regimes    |
> | Memorization evaluation        | Added generation-based memorization metrics     | Demonstrate practical benefits beyond loss metrics |
> | Theoretical foundation         | Integrated simplified theorem with empirical validation | Connect theory to observed scaling behavior      |
> | Training details transparency  | Added explicit token counts and training setup tables  | Clarify experimental conditions and reproducibility |
>
> We acknowledge that more work remains to fully validate SeqTD across the full spectrum of model scales and datasets. However, our new experiments provide compelling evidence that the method's benefits increase with scale, and the theoretical foundations help explain why. We look forward to your suggestions for additional experiments or analyses that would further address your concerns.

---

> > ### Comment · Reviewer_iFmB · 2025-04-03
> >
> > I have raised my score to 3, but I am really on the borderline. I do need to see I would prefer seeing runs at a higher parameter count closer to 1B. However, I understand this is difficult given potential compute restrictions.

---

> > > ### Author Response · Authors · 2025-04-09
> > >
> > > Thank you for acknowledging our new experiments. As far as scaling to larger model sizes is concerned, as of now we have the [following promising scaling trend](https://docs.google.com/presentation/d/1fiUtDQj2oTERn7KmHY8c20eX3WRi1Oyu27EH9DRnSvo/edit#slide=id.g347337d72f5_1_6) to justify why investing in SeqTD intervention at even larger scales will be worthwhile. The scaling of model parameters results in both reduction in loss of model utility, and decrease in memorization. While we would absolutely love to expand the parameter count of experiments even further, our current compute restricts this scope. We hope the scaling trends allow you to feel optimistic about being a stronger champion of this analysis-oriented work, with the goal of disentangling memorization and generalization in LLMs :)
> > >
> > > Thanks for your time!

---

### Official Review · Reviewer_hkbC · 2025-03-13

**Overall Recommendation:** 3

**Summary:**

The paper presents an investigation into sequence memorization in large language models (LLMs) and introduces Sequence-Tied Dropout (SeqTD) as a novel method to isolate memorization while maintaining generalization capabilities. The key argument is that typical memorization is not confined to specific neurons in standard training, making it difficult to remove without affecting overall model performance. The proposed SeqTD approach mitigates this issue by partitioning neurons into shared and memorization-specific groups, ensuring that memorization accumulates in a fixed subset of neurons while allowing general knowledge to remain broadly distributed.

**Claims And Evidence:**

The claims made in the paper are generally well-supported:

*  The claim that memorization is entangled with generalization in standard training is backed by experiments showing that removing neurons responsible for memorization also degrades model performance.
*   The assertion that SeqTD enables controlled memorization isolation is demonstrated through experiments where SeqTD effectively removes memorization with minimal performance loss.
*   The paper provides empirical evidence to support the claim that memorization neurons accumulate sequence-specific information while shared neurons maintain generalization.
*   However, Figure 4(a) presents some inconsistencies where validation loss without repetition is lower than with repetition. This result seems counterintuitive. The authors should clarify this discrepancy.

**Essential References Not Discussed:**

N/A

**Experimental Designs Or Analyses:**

The experimental design is generally solid, but some aspects could be improved:
* The results in Figure 4(a) are somewhat inconsistent with expected behavior; typically, repeated sequences should reinforce learning, but the validation loss is lower without repetition. This raises questions about whether the small-scale training setup accurately reflects real-world LLM behavior.
* Certain aspects need more discussion such as with SeqTD, the mem loss goes down and then up, this is not the case with standard training, what is the mechanism that drives this. Since early stopping with SeqTD could be detrimental to the privacy.

**Methods And Evaluation Criteria:**

The methods used are well-aligned with the problem:

* The experiments use TinyStories, TS-Repetition dataset, a reasonable choice for studying memorization in small-scale settings.
*    The evaluation metrics, including sequence forgetting and validation loss before and after neuron dropout, appropriately measure the effectiveness of SeqTD.
* The comparison with existing methods such as gradient attribution and pruning strengthens the validity of the results.
*  However, the experimental setup may not fully capture how SeqTD would behave in larger-scale LLMs, but the for the stated goal of the paper the choice of model and training suffices.

**Other Comments Or Suggestions:**

Typo
-  line 235 should use ` in latex rather than ' (i.e. open quote)

**Other Strengths And Weaknesses:**

Strengths

* The proposed SeqTD method is simple and novel (always a great combination) and well-supported empirical evidence.
* The methodology is rigorous and the ablation studies add depth to the evaluation.

Weaknesses
* The results in Figure 4(a) raise concerns about whether the small-scale training setup accurately represents LLM behavior.
* The forgetting evaluation could be more detailed beyond just loss increase. For example perplexity based ranks could be provided with and without the mitigation strategy.
* Results also suggest that training without repetition has better performance for memorized examples compared to SeqTD, this there is a gap to potential upper limit
* Certain aspects need more discussion such as with SeqTD, the mem loss goes down and then up, this is not the case with standard training, what is the mechanism that drives this. Since early stopping with SeqTD could be detrimental to the privacy.

**Questions For Authors:**

1. Figure 4(a) shows that validation loss without repetition is lower than with repetition. Can you clarify why this happens?

This seems inconsistent. If the small-scale setup does not reflect behavior in large-scale LLMs, that should be discussed explicitly. Please clarify this result.

2. Can the authors discuss why with SeqTD, the mem loss goes down and then up, this is not the case with standard training, what is the mechanism that drives this?

**Relation To Broader Scientific Literature:**

Most of the relevant memorization localization papers are included

**Theoretical Claims:**

There are no theory claims in the main paper.

---

> ### Author Rebuttal · Authors · 2025-04-01
>
> Thank you for the detailed and insightful feedback. We really appreciate that you found:
> - (i) SeqTD to be a **simple yet novel idea**,
> - (ii) the **methodology rigorous**, and
> - (iii) our **ablations helpful and revealing**.
>
> We've made specific additions to respond to your concerns.
>
> ---
> ### **Theme 1: Figure 4 (a) Validation Loss Dynamics**
> > *"Figure 4(a) shows that validation loss without repetition is lower than with repetition. Can you clarify why this happens?"*
>
> 🔗 *Full results available here:* [Token Scaling Experiment Results](https://shorturl.at/Aa7VZ)
>
> You've identified a crucial insight. In our initial setup, we treated data intervention (deduplication) as separate from model intervention (SeqTD), assuming only model interventions were appropriate for our analysis of disentangling memorization and generalization in LLMs. In Figure 4(a), the "without repetition" line represents an oracle baseline that involves direct data manipulation (deduplication). We initially considered this an "unfair" comparison because:
>
> - Our research focuses specifically on model-based interventions that can be applied without modifying training data, in order to understand the dynamics of generalization and memorization better.
> - Data deduplication was hence considered an oracle that represents an ideal but unreachable solution for the minimum validation loss on the trainig distribution.
>
> To address this ambiguity, we've conducted new experiments with a more realistic setting where repetition is actually beneficial to generalization:
>
> - We constructed a mixed dataset where 99% of tokens are from SlimPajama and 1% from TinyStories
> - The objective is to achieve optimal validation loss on TinyStories while minimizing memorization
> - In this realistic setting, some repetition is actually helpful (unlike our previous setup)
> - The new "validation optimal" occurs when TinyStories data is upsampled 10x (11M tokens)
>
> In the rebuttal [slides](https://shorturl.at/DqABg), we show in this setting, SeqTD significantly exceeds the deduplication baseline (reducing performance gap by 4x), while also reducing the loss gap between memorized and validation examples by 2.5x. This demonstrates SeqTD's capability to enable learning from repeated examples while mitigating memorization.
>
> ### **Theme 2: Figure 4 (b) "dip-and-rise" pattern in memorization loss**
> > Regarding the "dip-and-rise" pattern in memorization loss with SeqTD:
>
> - This can be explained by the conceptual intuition for SeqTD (Section 5.3) There is initially some memorization learned in the shared neurons early in training. Once the memorization neurons fit the repeated sequences, further gradient steps no longer reinforce memorization in the generalizing neurons. At this stage, the interference from other sequences forgets the memorization learned in the shared neurons -- resulting in the rising memorization loss.
> Importantly, our larger token scale training [results](https://shorturl.at/B51HI) do not exhibit this behavior, suggesting it may be eliminated when generalization neurons don't have the capacity to memorize sequences on their own.
>
> ---
> ### **Theme 3: Evaluation Clarity and Metric Expansion**
> > *"The forgetting evaluation could be more detailed beyond just loss increase."*
>
> 🔗 *Full results available here:*  [Google Slides – Memorization Metrics](https://shorturl.at/M7UfM)
>
> We've enhanced our evaluation with new results:
>
> - Perplexity-based ranks for memorized examples with and without SeqTD.
> - Memorization token accuracy metrics to complement loss measurements
> - Clear comparisons between SeqTD, standard training, and deduplication baselines
>
> These additions provide multiple perspectives on the effectiveness of SeqTD beyond simple loss measurements. The observed patterns in general follow the same trends as seen with loss based metrics.
>
> ### **Theme 4: Scaling experiments**
> While not a key concern in your review, we have developed new scaling trends and experiments based on other reviewer comments. If this is of interest, you can read our respons to Reviewer hZ7x.
>
> ---
> **Table 3: Summary of Actions for Reviewer hkbC**
> | Concern                         | Action Taken                                          | Expected Outcome                          |
> |----------------------------------|-------------------------------------------------------|-------------------------------------------|
> | Figure 4 (a) ambiguity               | Added larger scale experiments where repetition is beneficial | Clarify observed loss behavior and utility of SeqTD |
> | Loss dynamics explanation        | Added conceptual mechanism behind rising mem loss | Explain "dip-and-rise" pattern in memorization loss |
> | Evaluation diversity             | Added perplexity ranks + token accuracy              | Strengthen metrics beyond loss            |
>
> We would appreciate any suggestions for additional metrics or visualizations you believe would further strengthen our paper.

---

> > ### Comment · Reviewer_hkbC · 2025-04-01
> >
> > Thank you for your response, based on the results, I have increased my score.

---

> > > ### Author Response · Authors · 2025-04-09
> > >
> > > Thank you for acknowledging our additional experiments and clarifications! Your feedback on Figure 4 was especially valuable, and we believe the newer revision of the same significantly clarifies our finding.
> > >
> > > Thanks for your time!

---

### Official Review · Reviewer_hZ7x · 2025-03-14

**Overall Recommendation:** 2

**Summary:**

This work tackles privacy risks in LLMs from memorizing repeated sequences. Current post-hoc neuron isolation methods fail for data entangled with general capabilities. The authors propose SeqTD, a training method splitting neurons into shared (generalization) and memorization groups. By activating fixed memorization neurons for repeated sequences and shielding shared ones, SeqTD enables precise removal of memorized content without performance loss.

**Claims And Evidence:**

The main problem that SeqTD addresses is the challenge of isolating and removing memorized sequences from language models while preserving their general capabilities (localization). Existing post-hoc methods prove ineffective or significantly degrade model performance when memorized sequences are statistically similar to the broader training distribution. SeqTD offers a training-time approach that successfully identifies and removes such problematic memorization while maintaining the model's overall generalization abilities, demonstrating superior performance compared to previous methods.

**Essential References Not Discussed:**

[1] is cited but more discussion is needed.


[1] Gradient Routing: Masking Gradients to Localize Computation in Neural Networks, arxiv 2024.

**Experimental Designs Or Analyses:**

Experiments are rigorous in controlled settings: repeated vs. atypical sequences, ablation studies on dropout ratios, and noise tolerance . However, the absence of benchmarks on larger modelsor real-world data limits practical insights. The comparison to post-hoc methods is a strength, but including more recent baselines would enhance relevance.

**Methods And Evaluation Criteria:**

The evaluation approach is comprehensive and effective, employing appropriate metrics to assess sequence forgetting and model preservation capabilitie. For the method part, the authors may need to provide more insights why pretraining is better than post-hoc. Besides, the authors may also want to have more analysis about the difference of their method and [1].


[1] Gradient Routing: Masking Gradients to Localize Computation in Neural Networks, arxiv 2024.

**Other Comments Or Suggestions:**

I suggest addressing several minor inconsistencies in the manuscript: standardize the formatting of quotation marks and italics, currently some of them are confusing; maintain consistency with hyphenated terms like "trade-off"/"tradeoff" and "pretraining"/"pre-training"; and convert rasterized figures to vector graphics to improve visual quality and readability.

**Other Strengths And Weaknesses:**

This is generally a new and valuable method with comprehensive evaluation for the presented model. However, several critical aspects are underdeveloped:
1. deeper insights into why the training-time approach outperforms post-hoc methods would strengthen the theoretical foundation;
2. scalability questions remain regarding larger models and diverse datasets beyond TinyStories, raising concerns about the method's applicability to production-scale LLMs;
3. the paper lacks sufficient intuition about how SeqTD effectively disentangles memorization from generalization capabilities, which are often deeply intertwined in LLMs. While the implementation details and empirical results are thoroughly presented, the underlying mechanisms and theoretical guarantees for successful disentanglement deserve more attention.

**Questions For Authors:**

What mechanisms explain how your training-time approach successfully disentangles memorization from generalization capabilities in language models, and how might these insights scale to larger models trained on diverse datasets?

**Relation To Broader Scientific Literature:**

This works seem to be a new approach for localization of memorization. It may help privacy research for achieving a better trade-off between unlearning and performance.

**Theoretical Claims:**

N/A

---

> ### Author Rebuttal · Authors · 2025-04-01
>
> Thank you for your constructive and balanced review. We're glad you found:
> - (i) the **formulation of SeqTD clear and valuable**,
> - (ii) our **experiments well-designed within the scope**, and
> - (iii) the **motivation around privacy and unlearning promising**.
>
> We've carefully addressed your main concerns through new experiments and clarifications.
>
> ---
> ### **Theme 1: Scalability and Real-World Applicability**
> > *"The absence of benchmarks on larger models or real-world data limits practical insights."*
>
> This is an excellent point. To address this concern, we've conducted **additional scaling experiments** across multiple model sizes and more realistic training regimes:
>
> 🔗 *Full results available here:* [Model Scaling Experiment Results](https://shorturl.at/md9cQ), [Token Scaling Experiment Results](https://shorturl.at/Aa7VZ)
>
> - **Model scaling experiments** across four model sizes: 32M, 51M, 170M, 344M parameters. Our results show that SeqTD mitigates memorization across all model sizes, but the benefits of SeqTD increase with scale (no model degradation at 344M parameters)
> - **Token Scaling Experiments**  We train a 350M parameter model on a larger-scale corpus of 1 billion tokens containing a mix of TinyStories and SlimPajama. Our results demonstrate that SeqTD mitigates memorization (reducing the memorization-validation loss gap by 2.5x) while outperforming deduplication.
> - **Cross-architecture validation** To ensure the approach generalizes beyond a single model family, we perform experiments on the SmolLM-2 model family which uses a Gated MLP in contrast to the GPT-2 models trained in our original submission.
> ---
> ### **Theme 2: Intuition for Disentanglement Mechanism**
> > *"The paper lacks sufficient intuition about how SeqTD disentangles memorization."*
> >
> Thank you for highlighting this critical gap in our explanation. We've significantly enhanced the paper with clearer intuition about how and why SeqTD works by elevating the core insight from our theory in Appendix E (page 14):
> > **Simplified Theorem (E.2):**
> > *When repeated sequences consistently activate a fixed subset of neurons (via sequence-tied dropout), shared neurons gradually forget these patterns due to interference from other examples, while memorization neurons retain them.*
> >
> This theorem (pg. 13) formalizes the key mechanism: memorization neurons (activated less frequently) are shielded from forgetting during pretraining and accumulate sequence memorization. Thus, SeqTD can control where memorization accumulates by controlling how often neurons are active in pretraining.
>
> ---
> ### **Theme 3: Comparison to Gradient Routing**
> > *"The authors may want to include more discussion on how SeqTD compares with Gradient Routing [1]."*
>
> We appreciate this suggestion and have a dedicated comparison section in Section 4. In particular we discuss:
> - **Empirical Comparison**: As shown in Section 4.1 (pg. 4-5), our experiments with gradient routing demonstrated that such approaches can hinder cross-sequence learning and don't fully isolate memorization, whereas SeqTD preserves general capabilities while enabling more effective post-hoc removal
> - **Mechanistic Differences**: As discussed in Section 4.1, there are two crucial differences between SeqTD and Gradient Routing. Firstly, SeqTD allows for a pool of shared neurons (that are updated by all sequences) and our empirical results show this is crucial for generalization across sequences. Secondly, the dropout performed by SeqTD disincentivizes co-adaptations between general and memorization neurons. In gradient routing, we empirically find such co-adaptations develop which leads to an additional drop in performance after removing neurons.
>
> ---
> ### **Theme 4: Miscellaneous Improvements**
> Thank you for your helpful suggestions about presentation consistency. We duly take note of all of them and they have been updated in our internal overleaf draft.
>
> ---
> **Table 2: Summary of Actions for Reviewer hZ7x**
> | Concern                        | Action Taken                                          | Expected Outcome                         |
> |--------------------------------|-------------------------------------------------------|------------------------------------------|
> | Lack of scaling results        | Added experiments on model and token scaling | Demonstrated improved effectiveness at scale |
> | Intuition for disentanglement | Added simplified theorem and integrated with empirical observation| Clearer mechanistic understanding |
> | Gradient Routing comparison    | Added detailed empirical comparison and discussion of mechanistic differences | Clarified relationship to prior work |
> | Presentation consistency | Standardized formatting, improved figures, fixed typos | Enhanced readability |
>
> Please let us know if these revisions strengthen the paper and address your concerns about scalability, theoretical foundations, and comparisons to prior work. We're grateful for your thoughtful feedback!

---

### Official Review · Reviewer_Q8Dm · 2025-03-14

**Overall Recommendation:** 3

**Summary:**

The paper introduces a training strategy called Sequence-Tied Dropout (SeqTD) for large language models that aims to isolate memorized sequences into a specific subset of neurons while still allowing the model to learn general language patterns. The authors argue that standard training causes memorization to be entangled with general knowledge, making post-hoc removal of sensitive or copyrighted text problematic. The proposed method enforces a consistent dropout mask for repeated sequences based on sequence IDs, which channels memorization signals into designated neurons. Experiments on a modified TinyStories dataset demonstrate that SeqTD can “unlearn” repeated sequences effectively without degrading overall model performance, and theoretical analyses are provided to support the observed dynamics.

**Claims And Evidence:**

Claims: The paper claims that (1) traditional training leads to entangled memorization and generalization, (2) post-hoc localization methods such as pruning and gradient attribution are insufficient for typical repeated sequences, and (3) SeqTD can isolate memorization into dedicated neurons with minimal impact on overall performance.

Evidence: The authors support their claims with experiments comparing validation loss and sequence loss under different settings and provide theoretical analysis to explain the learning/forgetting dynamics. However, while the evidence is convincing for the controlled TinyStories setup, the experiments are limited to a single, small-scale dataset and do not explore more challenging benchmarks.

**Essential References Not Discussed:**

the literature review could be expanded to incorporate recent studies in copyright compliance and evaluation frameworks, such as:

1. “Copyright Violations and Large Language Models”
1. “Foundation Models and Fair Use”
1.  “Evaluating Copyright Takedown Methods for Language Models”
1. “LLMs and Memorization: On Quality and Specificity of Copyright Compliance”
1. “SHIELD: Evaluation and Defense Strategies for Copyright Compliance in LLM Text Generation”
1. “Digger: Detecting Copyright Content Misusage in Large Language Model Training”
1. “Speak, Memory: An Archaeology of Books Known to ChatGPT/GPT-4”
1. “Avoiding Copyright Infringement via Large Language Model Unlearning”
1. “CopyBench: Measuring Literal and Non-Literal Reproduction of Copyright-Protected Text in Language Model Generation”
1. “Preventing Verbatim Memorization in Language Models Gives a False Sense of Privacy”

Incorporating these references would provide a broader context and strengthen the paper’s discussion regarding practical applications and limitations in copyright-sensitive domains.

**Experimental Designs Or Analyses:**

The experiments on TinyStories illustrate the core ideas of SeqTD. However, the experimental design is limited in scope; testing on only a small-scale dataset does not fully demonstrate the method’s scalability or robustness in real-world applications.

Additionally, more thorough comparisons with alternative methods (beyond the baseline post-hoc localization techniques) would help in assessing the relative strengths and weaknesses of SeqTD.

**Methods And Evaluation Criteria:**

The methodology of partitioning MLP neurons into shared and memorization pools and enforcing a consistent dropout mask per sequence is clearly described.

The evaluation focuses on two key criteria: the loss increase on repeated sequences (indicating successful unlearning) and the validation loss (indicating generalization).

Although these metrics are reasonable, the evaluation would benefit from incorporating additional standardized benchmarks—such as those similar to cotaeval—to more comprehensively assess the impact on copyright compliance and overall performance.

**Other Comments Or Suggestions:**

It would be beneficial to include experiments on larger datasets or more realistic benchmarks to validate the general applicability of SeqTD.

Consider incorporating a standardized evaluation package (e.g., cotaeval or similar ones) to strengthen the experimental analysis.

Expanding the literature review to discuss related works in copyright evaluation and unlearning methods would improve the context and relevance of the paper.

**Other Strengths And Weaknesses:**

Strengths:

- The paper proposes a novel and conceptually clear method for disentangling memorization from general model capabilities.
- The theoretical analysis supports the empirical findings, and the idea of sequence-specific dropout is interesting and innovative in this context.

Weaknesses:

- The experimental evaluation is limited to a small, controlled dataset (TinyStories) and does not test scalability or robustness in more realistic settings.
- The literature review could be expanded to better position the work within recent developments in copyright compliance and related evaluation methods.

**Questions For Authors:**

See above

**Relation To Broader Scientific Literature:**

The paper is well-situated within the literature on memorization in large language models and unlearning methods. It addresses issues raised in works on model memorization and the challenges of removing sensitive content from pre-trained models.

**Theoretical Claims:**

The paper provides a series of theoretical results (e.g., Theorems E.1–E.3) to formalize the dynamics of memorization and forgetting under standard training and SeqTD.
While the proofs appear sound under the stated assumptions, a more detailed review of the derivations is necessary to fully validate the claims. No major issues were detected, but additional discussion on the assumptions’ practicality in real-world scenarios would strengthen the work.

---

> ### Author Rebuttal · Authors · 2025-04-01
>
> Thank you for your thoughtful and constructive review. We're glad that:
> - (i) you found our **theoretical analysis sound**,
> - (ii) appreciated the **clarity and motivation of our experiments**, and
> - (iii) recognized the novelty of using dropout for disentangling memorization from generalization.
>
> We've added new experiments and clarifications that we believe directly address your suggestions.
>
> ---
> ### **Theme 1: Experimental Scope & Generalization**
> > *"Experiments are limited to a small-scale dataset and do not explore more challenging benchmarks."*
>
> To expand the scope and test robustness, we've added:
>
> 🔗 *Full results available here:* [Model Scaling Experiment Results](https://shorturl.at/md9cQ), [Token Scaling Experiment Results](https://shorturl.at/Aa7VZ),  [Metrics Besides Loss](https://shorturl.at/M7UfM)
>
> - **Model scaling experiments** across four model sizes: 32, 51, 170, 344 million parameters. Our results show that SeqTD mitigates memorization across all model sizes but the benefits of SeqTD increase with scale (no model degradation at 344M parameters).
> - **Token Scaling Experiments**  We train a 350M parameter model on a larger-scale corpus of 1 billion tokens containing a mix of TinyStories and SlimPajama. Our results demonstrate that SeqTD mitigates memorization (reducing the memorization-validation loss gap by 2.5x) while outperforming a data-deduplication baseline on validation loss.
> - **Metrics Besides Loss** We validate SeqTD using generation-based extraction metrics of memorized sequence *token accuracy* and *memorized sequence perplexity rank*. We demonstrate SeqTD drops token accuracy on memorized sequences from >90%  to <10%, suggesting robust mitigation of memorization.
>
>
> ---
> ### **Theme 2: Clarifying Mechanism via Theory**
> > *"More practical insights into the theoretical assumptions."*
>
>  We've surfaced the core intuition behind our theoretical results through the following simplified theorem (based on the theory introduced in Appendix E, page 14)
>
> > **Simplified Theorem (E.3):**
> > *When a set of memorization neurons is activated consistently across repetitions of a given sequence and less frequently on other sequences, sequence memorization accumulates in these neurons and away from neurons activated on all sequences (shared neurons).*
>
> The key idea is that memorization neurons (by virtue of being less frequently activated) are shielded from interference from other gradient updates. As a result sequence memorization accumulates and is preserved in these neurons. On the other hand, memorization stored in the generalization neurons experiences forgetting dynamics and is eliminated as a result. **We empirically verify this proposed mechanism in Figure 6 of our paper by showing that SeqTD experiences smaller amplitude learning-forgetting cycles.*
>
> ---
> ### **Theme 3: Clarifying Evaluation Objectives (CotaEval)**
> > *"Evaluation would benefit from using standardized copyright benchmarks like cotaeval."*
>
> Thank you for the suggestion. While we agree with the **long-term motivation around copyright and privacy**, we want to clarify that this paper is a **foundational investigation into training dynamics**, not a compliance benchmark. The datasets used (e.g., TinyStories) are synthetic or constructed for studying memorization patterns—not real-world content where cotaeval would apply.
>
>
> ---
> ### **Theme 4: Literature Review and Broader Context**
> > *"Expand discussion to include recent works on copyright compliance."*
>
> We agree and have expanded our Related Work section in our internal draft to include the copyright compliance literature you suggested. Thanks for the suggestion!
>
>
> ---
> **Table 1: Summary of Actions for Reviewer Q8Dm**
> | Concern                      | Action Taken                                             | Expected Outcome                              |
> |------------------------------|----------------------------------------------------------|------------------------------------------------|
> | Small-scale experiment scope | Added model-size sweep + 1B-token training               | Demonstrate scaling properties and robustness         |
> | Evaluation depth             | Added perplexity rank + token accuracy metrics           | Provide stronger insights on model behavior    |
> | Mechanism clarity            | Moved and Simplified Theorem E.3             | Integrate theoretical foundation with empirical observations     |
> | CotaEval request             | Respectfully deferred; explained paper focus             | Clarified evaluation scope and contribution    |
> | Literature review gaps       | Added broader copyright/unlearning papers                | Situate paper in practical context             |
>
> We believe these additions substantially strengthen the paper while maintaining its focus on the core theoretical and empirical contributions. We appreciate your thoughtful suggestions that helped us improve the work's clarity, scope, and connection to the broader literature.

---

### Decision · Program_Chairs · 2025-05-01

**Decision:**

Accept (poster)

**Comment:**

This paper studies the question of identifying neurons responsible for memorization in LLMs.
The paper shows that standard training typically does not allow for memorization to be disentangled from model capabilities, and so removing neurons responsible for memorization also hurts utility.
The paper thus suggests a training recipe to force a more clear separation between neurons responsible for memorization and neurons responsible for generalization.

Reviewers found the idea interesting, but raised some concerns about the limited experimental scale, as well as a lack of justification for how the proposed method actually promotes disentanglement of neurons between memorization and generalization.
Despite these shortcomings, I think the paper presents an interesting new idea, which can lead to follow-up work on better understanding the learning dynamics of memorization.